# Q-LEARNING WITH POSTERIOR SAMPLING

**Priyank Agrawal & Shipra Agrawal**
Department of Industrial Engineering and Operations Research
Columbia University
{pa2608,sa3305}@columbia.edu

**Azmat Azati**[*]
BNP Paribas
{aa5288}@columbia.edu

## ABSTRACT

Bayesian posterior sampling techniques have demonstrated superior empirical performance in many exploration-exploitation settings. However, their theoretical analysis remains a challenge, especially in complex settings like reinforcement learning. In this paper, we introduce Q-Learning with Posterior Sampling (PSQL), a simple Q-learning-based algorithm that uses Gaussian posteriors on Q-values for exploration, akin to the popular Thompson Sampling algorithm in the multi-armed bandit setting. We show that in the tabular episodic MDP setting, PSQL achieves a regret bound of $\tilde{O}(H^2\sqrt{SAT})$, closely matching the known lower bound of $\Omega(H\sqrt{SAT})$. Here, S, A denote the number of states and actions in the underlying Markov Decision Process (MDP), and $T = KH$ with $K$ being the number of episodes and $H$ being the planning horizon. Our work provides several new technical insights into the core challenges in combining posterior sampling with dynamic programming and TD-learning-based RL algorithms, along with novel ideas for resolving those difficulties. We hope this will form a starting point for analyzing this efficient and important algorithmic technique in even more complex RL settings.

## 1 INTRODUCTION

In an online Reinforcement Learning (RL) problem, an agent interacts sequentially with an unknown environment and uses the observed outcomes to learn an interaction strategy. The underlying mathematical model for RL is a Markov Decision Process (MDP). In the tabular episodic setting, the MDP has a finite state space $\mathcal{S}$, a finite action space $\mathcal{A}$ and a planning horizon $H$. On taking an action $a$ in state $s$ at step $h$, the environment produces a reward and next state from the (unknown) reward model $R_h(s,a)$ and transition probability model $P_h(s,a)$ of the underlying MDP.

Q-learning (Watkins & Dayan, 1992) is a classic dynamic programming (DP)-based algorithm for RL. The DP equations (aka Bellman equations) provide a recursive expression for the optimal expected reward achievable from any state and action of the MDP, aka the $Q$-values, in terms of the optimal value achievable in the next state. Specifically, for any given $s \in \mathcal{S}, a \in \mathcal{A}, h \in [H]$, the Q-value $Q_h(s,a)$ is given by:

$$Q_h(s,a) = \max_{a \in \mathcal{A}} R_h(s,a) + \sum_{s' \in \mathcal{S}} P_h(s,a,s')V_{h+1}(s'), \text{ with} \tag{1}$$

$$V_{h+1}(s') := \max_{a' \in \mathcal{A}} Q_{h+1}(s',a'),$$

with $V_{H+1}(s) = 0, \forall s$. The optimal action in state $s$ is then given by the argmax action in the above.

When the reward and transition models of the MDP are unknown, the $Q$-learning algorithm uses the celebrated Temporal Difference (TD) learning idea (Sutton, 1988) to construct increasingly accurate estimates of $Q$-values using past observations. The key idea here is to construct an estimate of the right hand side of the Bellman equation, aka *target*, by *bootstrapping* the current estimate $\widehat{V}_{h+1}$ for the next step value function. That is, on playing an action $a$ in state $s$ at step $h$, and observing reward $r_h$ and next state $s'$, the target $z$ is typically constructed as: $z := r_h + \widehat{V}_{h+1}(s')$.

---

[*]work done as graduate student at Columbia

And the estimate $\widehat{Q}_h(s,a)$ for $Q_h(s,a)$ is updated to fit the Bellman equations using the *Q-learning update rule*[1]

$$\widehat{Q}_h(s,a) \leftarrow (1 - \alpha_n)\widehat{Q}_h(s,a) + \alpha_n z. \qquad (2)$$

Here $\alpha_n$ is an important parameter of the $Q$-learning algorithm, referred to as the learning rate. It is typically a function of the number of previous visits $n$ for the state $s$ and action $a$.

There are several ways to interpret the Q-learning update rule. The traditional frequentist interpretation popularized by Mnih et al. (2015) interprets this update as a gradient descent step for a least squares regression error minimization problem. We propose a more insightful interpretation of $Q$-learning obtained using Bayesian inference theory (details in Section 3.1). Specifically, if we assume a Gaussian prior

$$\mathcal{N}(\widehat{Q}_h(s,a), \tfrac{\sigma^2}{n-1})$$

on the $Q$-value $Q_h(s,a)$, and a Gaussian likelihood function $\mathcal{N}(Q_h(s,a), \sigma^2)$ for the target $z$, then using Bayes rule, one can derive the Bayesian posterior as the Gaussian distribution

$$\mathcal{N}(\widehat{Q}_h(s,a), \tfrac{\sigma^2}{n}), \qquad \text{where } \widehat{Q}_h(s,a) \leftarrow (1 - \alpha_n)\widehat{Q}_h(s,a) + \alpha_n z \qquad (3)$$

Importantly, the Bayesian posterior tracks not just the mean but also the variance or uncertainty in the $Q$-value estimate. Intuitively, the state and actions with a small number of past visits (i.e., small $n$) have large uncertainty in their current $Q$-value estimate, and should be explored more. The posterior sampling approaches implement this idea by simply taking a sample from the posterior, which is likely to be closer to the mean (less exploration) for actions with small posterior variance, and away from the mean (more exploration) for those with large variance. This uncertainty quantification is useful for managing the exploration-exploitation tradeoff for regret minimization. The exploration methodology is distinct from algorithms that use additive bonuses or randomized perturbations in the estimates.

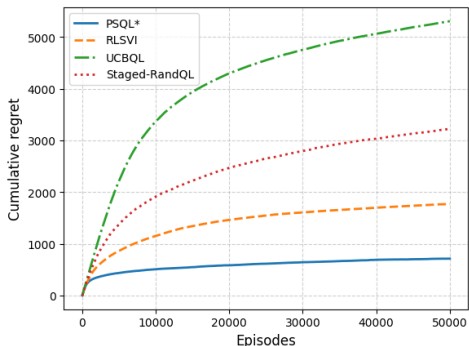

Figure 1: Performance comparison of PSQL*(a heuristic derived from PSQL), UCBQL (Jin et al., 2018); Staged-RandQL(Tiapkin et al., 2023), and RLSVI (Russo, 2019) in a chain MDP environment (for details, and more experiments, see Appendix A).

Following this intuition, we introduce Q-learning with Posterior Sampling (PSQL) algorithm that maintains a posterior on Q-values for every state and action. Then, to decide an action in any given state, it simply generates a sample from the posterior for each action, and plays the arg max action of the sampled $Q$-values.

Popularized by their success in the multi-armed bandit settings (Thompson, 1933; Chapelle & Li, 2011; Kaufmann et al., 2012; Agrawal & Goyal, 2017), and in deep reinforcement learning regimes (Osband et al., 2016a; Fortunato et al., 2017; Azizzadenesheli et al., 2018; Li et al., 2021b; Fan & Ming, 2021; Sasso et al., 2023), the posterior sampling approaches are generally believed to be more efficient in managing the exploration-exploitation tradeoff than their UCB (Upper Confidence Bound) counterparts. Our preliminary experiments (see Figure 1) suggest that this is also the case for our Q-learning approach in the tabular RL setting. [2] However, obtaining provable guarantees for posterior sampling approaches have historically been more challenging.

Several previous works (e.g., Li et al. (2021a); Jin et al. (2018)) use UCB-based exploration bonuses to design optimistic Q-learning algorithms with near-optimal regret bounds.[3]. For the posterior sampling based approaches however, the first tractable

---

[1]Or, $\widehat{Q}_h(s,a) \leftarrow \widehat{Q}_h(s,a) + \alpha_n(z - \widehat{Q}_h(s,a))$ where $z - \widehat{Q}_h(s,a)$ is called the Temporal Difference (TD).

[2]For the empirical study reported here, we implement a vanilla version of posterior sampling with Q-learning. The PSQL algorithm presented later modifies the target computation as described later in Section 3, for the sake of theoretical analysis.

[3]The algorithm from Jin et al. (2018) is referred as UCBQL in the text and experiments

$Q$-learning based algorithm with provable regret bounds was provided only recently by Tiapkin et al. (2023) for the Staged-RandQL algorithm. However, the Staged-RandQL (and RandQL) algorithm presented in their work deviated from the natural approach of putting a posterior on Q-values, and instead derived a Dirichlet Bayesian posterior on the transition probabilities, which is conceptually closer to some model-based posterior sampling algorithms, e.g., the PSRL algorithm in Agrawal & Jia (2017). RandQL implements sampling from the implied distribution on Q-estimates in a more efficient way via learning rate randomization, so that it qualifies as a model-free algorithm.

Another closely related approach with provable regret bounds is the RLSVI (Randomized Least Squared Value Iteration) algorithm by Osband et al. (2016b; 2019); Russo (2019); Zanette et al. (2020); Agrawal et al. (2021); Xiong et al. (2022). The RLSVI algorithm is an approximate value iteration-based approach that can be interpreted as maintaining "empirical posteriors" over the value functions by injecting noise. However, in the tabular setting considered here, RLSVI reduces to a model-based algorithm. (See Section 2.1 for further comparisons.) There are several other model-based posterior sampling algorithms in the literature with near optimal regret bounds (Osband et al., 2013; Osband & Van Roy, 2017; Ouyang et al., 2017; Agrawal & Jia, 2022; Agarwal & Zhang, 2022; Tiapkin et al., 2022). Model-based algorithms directly estimate the reward and/or transition model, instead of the implied optimal value functions, or policy parameters. In many settings, model-based algorithms can be more sample efficient. But, model-free approaches like Q-learning have gained popularity in practice because of their simplicity and flexibility, and underlie most successful modern deep RL algorithms (e.g., DQN Mnih et al. (2013), DDQN van Hasselt et al. (2015), A3C Mnih et al. (2016)). Provable regret bounds for a simple posterior sampling based Q-learning algorithm like PSQL, therefore, still remains a problem of significant interest.

Our contributions are summarized as follows.

- We propose the Q-learning with Posterior Sampling (PSQL) algorithm that is the *first* Q-learning algorithm with natural and efficient exploration provided by the Bayesian posterior sampling approach. Our preliminary experiments demonstrate promising empirical performance of this simple algorithm compared to contemporary approaches. (See Section 3 for algorithm design and Appendix A for experiments.)

- We provide a novel derivation of Q-learning as a solution to a Bayesian inference problem with a regularized Evidence Lower Bound (ELBO) objective. Besides forming the basis of our PSQL algorithm design, this derivation provides a more insightful interpretation of the learning rates introduced in some previous works on $Q$-learning (e.g., Jin et al. (2018)) to obtain provable regret bounds. (See Section 3.1.)

- We prove a near-optimal regret bound of $\tilde{O}(H^2\sqrt{SAT})$ for PSQL which closely match the known lower bound of $\Omega(H\sqrt{SAT})$ (Jin et al., 2018). Our result improves the regret bounds available for the closely related approach of RLSVI (Russo, 2019) and matches those recently derived by Tiapkin et al. (2023) for a more complex posterior sampling based algorithm Staged-RandQL. (See Section 2, 4.)

- Our regret analysis reveals several key difficulties in combining posterior sampling with DP and TD-learning-based algorithms due to error accumulation in the bootstrapped target; along with novel ideas for overcoming these challenges. (See Section 4.1.)

## 2 OUR SETTING AND MAIN RESULT

In the online reinforcement learning setting, the algorithm interacts with environment in $K$ sequential episodes, each containing $H$ steps. At step $h = 1, \ldots, H$ of each episode $k$, the algorithm observes the current state $s_{k,h}$, takes an action $a_{k,h}$ and observes a reward $r_{k,h}$ and the next state $s_{k,h+1}$.

The reward and next state are generated by the environment according to a fixed underlying MDP $(\mathcal{S}, \mathcal{A}, R, P)$, so that $\Pr(s_{k,h+1} = s'|s_{k,h} = s, a_{k,h} = a) = P_h(s, a, s'), \mathbb{E}[r_{k,h}|s_{k,h} = s, a_{k,h} = a] = R_h(s, a)$. However, the reward functions and the transition probability distributions $R_h, P_h, h = 1, \ldots, H$ are apriori unknown to the algorithm. The goal is to minimize total regret compared to the optimal value given by the dynamic programming equation (1). Specifically, let $\pi_k$ denote the policy used by the algorithm in episode $k$, so that $a_{k,h} = \pi_k(s_{k,h})$. We aim to bound regret, defined as

$$\text{Reg}(K) := \sum_{k=1}^{K}(V_1(s_{k,1}) - V^{\pi_k}(s_{k,1})) \tag{4}$$

for any set of starting states $s_{k,1}, k = 1, \ldots, K$.

Since the algorithm can only observe the environment's response at a visited state and action, a main challenge in this problem is managing the exploration-exploitation tradeoff. This refers to the dilemma between picking actions that are most likely to be optimal according to the observations made so far, versus picking actions for that allow visiting less-explored states and actions. The two main approaches for managing exploration-exploitation tradeoffs are the optimistic approaches based on UCB and posterior sampling approaches (aka Thompson Sampling in multi-armed bandit settings).

In this paper, we present a Q-learning algorithm with posterior sampling (PSQL) that achieves the following regret bound. Here $\tilde{O}(\cdot)$ hides absolute constants and logarithmic factors.

**Theorem 1** (Informal). *The cumulative regret of our PSQL (Algorithm 1 ) in $K$ episodes with horizon $H$ is bounded as $Reg(K) \leq \tilde{O}\left(H^2\sqrt{SAT}\right)$, where $T = KH$.*

## 2.1 RELATED WORKS

| | Algorithm | Regret | Comments |
|---|---|---|---|
| UCB | UCBQL (Jin et al., 2018) | $\tilde{O}(H^{1.5}\sqrt{SAT})$ | Q-learning with UCB |
| | Q-EarlySettled-Advantage (Li et al., 2021a) | $\tilde{O}(H\sqrt{SAT})$ | Q-learning with UCB |
| Posterior sampl. | Conditional Posterior Sampling (Dann et al., 2021) | $\tilde{O}(HSA\sqrt{T})$ | computationally intractable |
| | RLSVI (Russo, 2019) | $\tilde{O}(H^3 S^{1.5}\sqrt{AT})$ | approximate value iteration |
| | C-RLSVI (Agrawal et al., 2021) | $\tilde{O}(H^2 S\sqrt{AT})$ | approximate value iteration |
| | Staged RandQL (Tiapkin et al., 2023) | $\tilde{O}(H^2\sqrt{SAT})$ | randomized learning-rates |
| | **PSQL** [this work] | $\tilde{O}(H^2\sqrt{SAT})$ | Gaussian posteriors on Q-values |
| | Lower bound (Jin et al., 2018) | $\Omega(H\sqrt{SAT})$ | - |

Table 1: Comparison of our regret bound to related works ( Dann et al. (2021) is in function approximation setting).

Our work falls under the umbrella of online episodic reinforcement learning on regret minimization in tabular setting. In the category of the Upper Confidence Bound (UCB)-based algorithms, there is a huge body of research both on model-based (Bartlett & Tewari, 2012; Azar et al., 2017; Fruit et al., 2018; Zanette & Brunskill, 2019; Zhang et al., 2020; Boone & Zhang, 2024), and model-free (Jin et al., 2018; Bai et al., 2019; Ménard et al., 2021; Zhang & Xie, 2023; Agrawal & Agrawal, 2024) algorithms. In-fact, Jin et al. (2018) were the first to provide a near-optimal worst-case regret bound of $\tilde{O}(\sqrt{H^3 SAT})$, subsequently improved to $\tilde{O}(H\sqrt{SAT})$ by Zhang et al. (2020); Li et al. (2021a).

Motivated by the superior empirical performance of Bayesian posterior sampling approaches compared to their UCB counterparts (Chapelle & Li, 2011; Kaufmann et al., 2012; Osband et al., 2013; Osband & Van Roy, 2017; Osband et al., 2019) there have been several attempts at deriving provable regret bounds for these approaches in the episodic RL setting. Among model-based approaches, near optimal regret bounds have been established for approaches that use (typically Dirichlet) posteriors on transition models (Ouyang et al., 2017; Agrawal & Jia, 2017; Tiapkin et al., 2022). There have been relatively limited studies on model-free, sample-efficient and computationally efficient Bayesian algorithms. Dann et al. (2021) proposed one such framework but is computationally intractable. *Our work aims to fill this gap.*

A popular approach closely related to posterior sampling is Randomized Least Square Value Iteration (RLSVI) (Osband et al., 2016b; 2019; Russo, 2019; Zanette et al., 2020) and RLSVI-like approaches (Agrawal et al., 2021; Xiong et al., 2022; Ishfaq et al., 2021; 2023; 2024). In RLSVI, the exploration is carried out by injecting randomized uncorrelated noise to the reward samples, followed by a re-fitting of a $Q$-function estimate by solving a least squares problem on *all the past data*, incurring heavy computation and storage costs. This process has been interpreted as forming an approximate posterior distribution over value functions. RLSVI too enjoys a "superior-than-UCB" empirical performance. In contrast to Q-learning (or TD-learning approaches in general) these approaches do not bootstrap on the older estimates and hence their techniques are not broadly applicable

in our analysis. However, its worst-case regret bounds (Russo, 2019) remain suboptimal (see Table 1) in their dependence on the size of the state space.

More recently, Tiapkin et al. (2023) proposed (RandQL and Staged-RandQL) algorithms that are model-free, tractable and enjoy $\tilde{O}(H^2\sqrt{SAT})$ regret by randomizing the learning rates of the Q-learning update rule. Their algorithmic design is based on Dirichlet posteriors on transition models and efficient implementation of the implied distribution on $Q$-value estimates via learning rate randomization. Our algorithm is much simpler with far lesser randomized sampling steps and in our preliminary experiments (see Figure 1 and Section A), our PSQL approach with simple Gaussian based posteriors shows better/comparable performance compared to these algorithms.

In Table 1 we provide a detailed comparison of our results with the above-mentioned related work on posterior sampling algorithms for RL.

## 3 ALGORITHM DESIGN

We first present a Bayesian posterior-based derivation of the Q-learning update rule, which forms the basis for our algorithm design.

### 3.1 POSTERIOR DERIVATION

An insightful interpretation of the Q-learning update rule can be obtained using Bayesian inference. Let $\theta$ denote the Bayesian parameter that we are inferring, which in our case is the quantity $Q_h(s,a)$. Given a prior $p$ on $\theta$, log likelihood function $\ell(\theta, \cdot)$, and a sample $z$, the Bayesian posterior $q$ is given by the Bayes rule:

$$q(\theta) \propto p(\theta) \cdot \exp(\ell(\theta, z)) \tag{5}$$

which can also be derived as an optimal solution of the following optimization problem (see Chapter 10 in Bishop & Nasrabadi (2006)), whose objective is commonly referred to as Evidence Lower Bound (ELBO):

$$\max_q \mathbb{E}_{\theta \sim q}[\ell(\theta, z)] - KL(q||p) \tag{6}$$

where $KL(\cdot||\cdot) = \int_\theta q(\theta) \log(\frac{q(\theta)}{p(\theta)})$ denotes KL-divergence function. It is well known that when $p(\cdot)$ is Gaussian, say $\mathcal{N}(\widehat{\mu}, \frac{\sigma^2}{n-1})$, and the likelihood given $\theta$ is Gaussian $\mathcal{N}(\theta, \sigma^2)$, then the posterior $q(\cdot)$ is given by the Gaussian distribution

$$\mathcal{N}(\widehat{\mu}, \frac{\sigma^2}{n}), \text{ with } \widehat{\mu} \leftarrow (1 - \alpha_n)\widehat{\mu} + \alpha_n z \tag{7}$$

with $\alpha_n = \frac{1}{n}$. Therefore, substituting $\theta$ as $Q_h(s,a)$ and $\widehat{\mu}$ as $\widehat{Q}_h(s,a)$, the above yields the Q-learning learning update rule (2) with learning rate $\alpha_n = \frac{1}{n}$.

A caveat is that the above assumes $z$ to be an unbiased sample from the target distribution, whereas in Q-learning, $z$ is biased due to bootstrapping. In a recent work, Jin et al. (2018) observed that in order to account for this bias and obtain theoretical guarantees for Q-learning, the learning rate needs to be adjusted to $\alpha_n = \frac{H+1}{H+n}$. In fact, Bayesian inference can also provide a meaningful interpretation of this modified learning rate proposed in Jin et al. (2018). Consider the following "regularized" Bayesian inference problem (Khan & Rue, 2023) which adds an entropy term to the ELBO objective in (6):

$$\max_q \mathbb{E}_{\theta \sim q}[\ell(\theta, z)] - KL(q||p) + \lambda_n \mathcal{H}(q) \tag{8}$$

where $\mathcal{H}(q)$ denotes the entropy of the posterior. We show in Lemma B.1 (refer Appendix B) that for the choice of $\lambda_n = \frac{H}{n}$, when the prior $p(\theta)$ is Gaussian $\mathcal{N}(\widehat{\mu}, \frac{\sigma^2}{n-1})$, and the likelihood of $z$ given $\theta$ is Gaussian $\mathcal{N}(\theta, \frac{\sigma^2}{H+1})$, then the posterior $q(\cdot)$ is given by the Gaussian distribution in (7) *with the same learning rate $\alpha_n = \frac{H+1}{H+n}$* as suggested in Jin et al. (2018).

Substituting $\theta$ as $Q_h(s,a)$ and $\widehat{\mu}$ has $\widehat{Q}_h(s,a)$ in (8), we derive that given a Gaussian prior $\mathcal{N}(\widehat{Q}_h(s,a), \frac{\sigma^2}{n-1})$ over $Q_h(s,a)$, Gaussian likelihood $\mathcal{N}(Q_h(s,a), \frac{\sigma^2}{H+1})$ on target $z$, the following posterior maximizes the regularized ELBO objective:

$$\mathcal{N}(\widehat{Q}_h(s,a), \frac{\sigma^2}{n}), \text{ where } \quad \widehat{Q}_h(s,a) \leftarrow (1 - \alpha_n)\widehat{Q}_h(s,a) + \alpha_n z, \tag{9}$$

where $\alpha_n = \frac{H+1}{H+n}$, and $n$ is the number of samples for $s, a$ observed so far.

The entropy regularization term of (8) introduces extra uncertainty in the posterior. Intuitively, this makes sense for Q-learning as the target $z$ is bootstrapped on previous interactions and likely has additional bias. The weight $\lambda_n$ of this entropy term decreases as the number of samples increases and the bootstrapped target is expected to have lower bias. This derivation may be of independent interest as it provides an intuitive explanation of the modified learning rate schedule proposed ($\frac{H+1}{H+n}$ as compared to $\frac{1}{n}$) in Jin et al. (2018), where it was motivated mainly by the mechanics of regret analysis. The above posterior derivation forms the basis of our algorithm design presented next.

### 3.2 Algorithm details

A detailed pseudo-code of our PSQL algorithm is provided as Algorithm 1,2. It uses the current Bayesian posterior to generate samples of $Q$- values at the current state and all actions, and plays the arg max action. Specifically, at a given episode $k$, let $s_h$ be the current state observed in the beginning of the episode and for action $a$, let $N_h(s_h, a)$ be the number of visits of state $s_h$ and action $a$ before this episode. Let $\widehat{Q}_h(s_h, a)$ be the current estimate of the posterior mean, and

$$\sigma(n) = \frac{\sigma^2}{n+1} := 64 \frac{H^3}{n+1} \log(KH/\delta). \tag{10}$$

Then, the algorithm samples for each $a$,

$$\tilde{Q}_h(s_h, a) \sim \mathcal{N}(\widehat{Q}_h(s_h, a), \sigma(N_h(s_h, a))^2)$$

and plays the arg max action $a_h := \arg\max_a \tilde{Q}_h(s_h, a)$.

The algorithm then observes a reward $r_h$ and the next state $s_{h+1}$, computes a target $z$, and updates the posterior mean estimate using the Q-learning update rule. A natural setting of the target would be $r_h + \max_{a'} \tilde{Q}_{h+1}(s_{h+1}, a')$, which we refer to as the "vanilla version" or PSQL*. However, due to unresolvable difficulties in regret analysis discussed later in Section 4, the PSQL algorithm computes the target in a slightly optimistic manner ($r_h + \overline{V}_{h+1}(s)$) as we describe later in this section. Our experiments (Appendix A) show that although this modification does impact performance, PSQL still remains significantly superior to its UCB counterpart.

---

**Algorithm 1** Q-learning with Posterior Sampling (PSQL)

1: **Initialize:** $\widehat{Q}_{H+1}(s, a) = \widehat{V}_{H+1}(s) = 0, \widehat{Q}_h(s, a) = \widehat{V}_h(s) = H, N_h(s, a) = 0 \, \forall s, a, h$.
2: **for** episodes $k = 1, 2, \ldots$ **do**
3:     Observe $s_1$.
4:     **for** step $h = 1, 2, \ldots, H$ **do**
5:         Sample $\tilde{Q}_h(s_h, a) \sim \mathcal{N}(\widehat{Q}_h(s_h, a), \sigma(N_h(s_h, a))^2)$, for all $a \in \mathcal{A}$.
6:         Play $a_h := \arg\max_{a \in \mathcal{A}} \tilde{Q}_h(s_h, a)$.
7:         Observe $r_h$ and $s_{h+1}$.
8:         $z \leftarrow \texttt{ConstructTarget}(h, r_h, s_{h+1}, \widehat{Q}_{h+1}, N_{h+1})$.
9:         $n := N_h(s_h, a_h) \leftarrow N_h(s_h, a_h) + 1, \alpha_n := \frac{H+1}{H+n}$.
10:        $\widehat{Q}_h(s_h, a_h) \leftarrow (1 - \alpha_n)\widehat{Q}_h(s_h, a_h) + \alpha_n z$.
11:     **end for**
12: **end for**

---

Specifically, given reward $r_h = r$ and next state $s_{h+1} = s'$, a value function estimate $\overline{V}_{h+1}(s')$ is computed as the maximum of $J$ samples from the posterior on $Q_{h+1}(s', \widehat{a})$, with $\widehat{a}$ being the arg max action of posterior mean + standard deviation. That is, let

$$\widehat{a} := \arg\max_a \widehat{Q}_{h+1}(s', a) + \sigma(N_{h+1}(s', a)), \text{ and } \overline{V}_{h+1}(s') = \max_{j \in J} \tilde{V}^j,$$

$$\text{with } J = J(\delta) := \frac{\log(SAT/\delta)}{\log(4/(4-p_1))}, \quad p_1 = \Phi(-1) - \frac{\delta}{H} - \delta. \tag{11}$$

Observe that the above procedure computes a $\overline{V}_{h+1}(s')$ that is more optimistic than single sample maximum (i.e., "vanilla version" $\max_{a'} \tilde{Q}_h(s', a')$). However, the optimism is limited only to the

---

**Algorithm 2** `ConstructTarget`$(h, r, s', \widehat{Q}, N)$

---

**Return** $r$, if $h = H + 1$.
Set $\widehat{a} = \arg\max_a \widehat{Q}(s', a) + \sigma(N(s', a))$. Set $J := J(\delta)$ as in (11).
/* Take maximum of the $J$ samples from the posterior of target $V_{h+1}$ */
Sample $\tilde{V}^j \sim \mathcal{N}\left(\widehat{Q}(s', \widehat{a}), \sigma(N(s', \widehat{a}))^2\right)$, for $j \in [J(\delta)]$, $a \in \mathcal{A}$.
$\overline{V}(s') \leftarrow \max_j \tilde{V}^j$.
**Return** $z := r + \overline{V}(s')$.

---

target computation not to the main decision-making in Line 6, marking an important departure from UCB based optimism (e.g., Jin et al. (2018)). Multiple sampling from the posteriors is a common technique considered in the past works (Tiapkin et al., 2022; Agrawal & Jia, 2017; Agrawal et al., 2017) to aid analysis. Finally, the algorithm uses the computed target $z = r_h + \overline{V}_{h+1}(s_{h+1})$ to update the posterior mean via the Q-learning update rule, with $\alpha_n = \frac{H+1}{H+n}$:

$$\widehat{Q}_h(s, a) \leftarrow (1 - \alpha_n)\widehat{Q}_h(s, a) + \alpha z.$$

Let $n = N_{k,h}(s, a)$, then Algorithm 1 implies ($\alpha_n^0 := \Pi_{j=1}^n (1 - \alpha_j)$ and $\alpha_n^i := \alpha_i \Pi_{j=i+1}^n (1 - \alpha_j)$),

$$\widehat{Q}_{k,h}(s, a) = \alpha_n^0 H + \sum_{i=1}^n \alpha_n^i \left(r_{k_i, h} + \overline{V}_{k_i, h+1}(s_{k_i, h+1})\right). \tag{12}$$

## 4 REGRET ANALYSIS

We prove the following regret bound for PSQL.

**Theorem 2.** *The cumulative regret of PSQL (Algorithm 1,2) in $K$ episodes satisfies*

$$Reg(K) := \left(\sum_{k=1}^K V_1^*(s_{k,1}) - V_1^{\pi_k}(s_{k,1})\right) \leq O\left(H^2\sqrt{SAT}\chi\right),$$

*with probability at least $1 - \delta$, where $\chi = \log(JSAT/\delta)$ and $T = KH$.*

### 4.1 CHALLENGES AND TECHNIQUES.

Most of the unique challenges for the theoretical analysis of Q-learning with posterior sampling are associated with the bootstrapped nature of TD-learning itself. As shown in (12), mean estimate at the given step $h$ depends on a weighted average of the past next-step $h + 1$ estimates, causing the errors at $h + 1$ of the past estimates propagate to the estimate at step $h$. In model-based methods(e.g., Azar et al. (2017); Osband & Van Roy (2017); Zanette et al. (2020)) such issues are non-existent as they recalculate their estimates from scratch at each time step.

**Optimism dies down under recursion.** One difficulty in analyzing Bayesian posterior sampling algorithms is the absence of high probability optimism (the property that the estimates upper bound the true parameters). Observe that the regret of an algorithm in any episode $k$ can be decomposed as:

$$V_1^*(s_{k,1}) - V_1^{\pi_k}(s_{k,1}) \leq \underbrace{(V_1^*(s_{k,1}) - \tilde{Q}_{k,1}(s_{k,1}, a_{k,1}))}_{\text{Optimism error}} + \underbrace{(\tilde{Q}_{k,1}(s_{k,1}, a_{k,1}) - V_1^{\pi_k}(s_{k,1}))}_{\text{Estimation error}}. \tag{13}$$

In algorithms like UCBQL (Jin et al., 2018), there is no optimism error since the UCB estimate is a high-confidence upper bound on the optimal value function. Prior posterior sampling approaches (Agrawal & Goyal, 2012; 2017; Russo, 2019; Agrawal et al., 2021) were able to bound optimism error by proving a constant probability optimism, and then boosting to high probability by a statistical argument. However, due to the recursive nature of Q-learning, their techniques do not directly apply.

To see this, suppose that if we have a constant probability $p$ of optimism of posteriors on value functions in state $H$. The optimism of stage $H - 1$ value requires optimism of stage $H$ value; leading to $p^2$ probability of optimism in stage $H - 1$. Continuing this way, we get an exponentially small probability of optimism for stage 1.

**Multiple sampling from the posteriors only partially helps.** To get around the issues with constant probability optimism, many posterior sampling algorithms (e.g., model-based PSRL Agrawal & Jia (2022), MNL-bandit Agrawal et al. (2017),Tiapkin et al. (2022),Ishfaq et al. (2021) etc.) taking max over multiple (say $J$) samples from the posterior in order to get high probability optimism. We follow a similar modification, with differences described later in the section. We believe that, due to bootstrapping nature of Q-learning (or TD-learning methods in general), merely taking multiple samples for either decision-making and the target construction would lead to exponential (in $H$) accumulation of errors, even for $J$ as small as 2. Below, we provide a rough argument.

We expect the bias of $\tilde{Q}_h$ (sample from the posterior distribution) to track $Q_h^*$ with error that scales as standard deviation of $\tilde{Q}_h$) (lets call that error as $\epsilon$). Now, suppose $J$ samples are used in decision-making (i.e., we take multiple samples from the posterior distributions at Line 6 of Algorithm 1). We incur regret whenever the bias of $\max_j \tilde{Q}^j$ exceeds $Q^*$. Using the standard techniques, this error at step $H$ has an error bound of $\epsilon\sqrt{J \log(1/\delta)}$ with probability $1 - \delta$. This error subsequently propagates multiplicatively via bootstrapping of Bellman equation. For the step $H - 1$, the optimism error contribution will be $\epsilon\sqrt{J^2 \log(1/\delta)}$ with probability $1 - \delta$. Continuing this argument, at step 1, the cumulative optimism error will be of the order $\epsilon\sqrt{J^H \log(1/\delta)}$, i.e., exponential in $H$ for any $J \geq 2$. The usual trick of obtaining high probability optimism by taking multiple samples from the posterior doesn't work for Q-learning, at least not without further novel ideas.

**Our techniques.** The design of target computation procedure is pivotal to PSQL. Our algorithm design is characterized by two key items: (1) using optimistic posterior sampling in target computation *only*; (2) using the argmax action $\widehat{a}$ of the posterior mean (with a standard deviation offset) in our target computation.

First is motivated by the observation that to break the recursive multiplicative decay in constant probability of optimism, we just need to ensure high probability optimism of the next-stage value function estimate used in the target computation. Second is motivated by the previous discussion that merely taking multiple samples may lead to exponential error. In our analysis, we show that the action $\widehat{a}$ is a special action, whose standard deviation is close to the the played action $a_h$ with constant probability (Lemma C.2). As a result, we are able to demonstrate that as in the standard Q-learning $\overline{V}_h(s_h)$ (defined with $\widehat{a}$) cannot be too far from $\tilde{Q}_h(s_h, a_h)$. Intuitively they are tracking the similar quantities. In summary, our algorithm uses a combination of vanilla (single-sample) and optimistic (multiple-sample) posterior sampling for action selection and target computation, respectively.

### 4.2 PROOF SKETCH

We provide a proof sketch for Theorem 2. All the missing details from this section are in Appendix C. Here, we use $\tilde{Q}_{k,h}, \widehat{Q}_{k,h}, \overline{V}_{k,h}, N_{k,h}$, to denote the values of $\tilde{Q}_h, \widehat{Q}_h, \overline{V}_h, N_h$, respectively at the beginning of episode $k$ of Algorithm 1, 2. And, as before, $s_{k,h}, a_{k,h}$ denote the state and action visited at episode $k$, step $h$.

Following the regret decomposition in (13) we bound the regret by bounding optimism error and estimation error. We introduce several new technical ideas to this end. Leveraging our algorithm design, we first prove that $\overline{V}_{k,h}$ is a tracking upper bound (optimistic estimate) to $V_h^*$. Second and the most crucial bit is to show that deviation of $\overline{V}_{k,h}(s_{k,h})$ from the sample used in decision-making, $\tilde{Q}_{k,h}(s_{k,h}, a_{k,h})$, can be tractably bounded across rounds of interactions. This combined with the optimism of $\overline{V}_{k,h}$, naturally bounds the *optimism error*. Third we demonstrate the estimation error has a recursive structure, i.e., error at step $h$ depends on error at $h + 1$ and terms attributed to stochasticity in the model; and deviation of $\overline{V}_{k,h}(s_{k,h})$ from $\tilde{Q}_{k,h}(s_{k,h}, a_{k,h})$. Therefore, first two parts are utilized to prove *estimation error* bound.

**(a) $\overline{V}_{k,h}$ used in the target, is an optimistic estimate of $V_h^*$**

**Lemma 1** (Abridged). *For any episode $k$ and index $h$, the following holds with probability at least $1 - \delta/KH$,*

$$\overline{V}_{k,h}(s_{k,h}) \geq V_h^*(s_{k,h}),$$

*where $\delta$ is a parameter of **PSQL** used to define the number of samples $J$ used to compute the target.*

The above is an abridged version of Lemma C.1. The proof of which inductively uses the optimism and estimation error bounds available for the next stage $(h + 1)$ to bound the estimation error in the posterior mean $\widehat{Q}_h(s, a)$. Then, anti-concentration (i.e., lower tail bounds) of the Gaussian posterior distribution provides the desired constant probability optimism.

**(b)** $\overline{V}_{k,h}(s_{k,h})$ **used in the target, is not far away from** $\tilde{Q}_{k,h}(s_{k,h}, a_{k,h})$**.** The following lemma is an abridged version of Lemma C.3 and tells us that the gap is of the order of $\sigma(N_{k,h}(s_{k,h}, a_{k,h}))$ which goes down (see (10)) as $s_{k,h}, a_{k,h}$ is visited often with rate $1/\sqrt{N_{k,h}(s_{k,h}, a_{k,h})}$. This is central to our analysis.

**Lemma 2** (Abridged). *In Algorithm 1, with probability $1 - 2\delta$, the following holds for all $k \in [K]$ and $h \in [H]$,*

$$\overline{V}_{k,h}(s_{k,h}) - \tilde{Q}_{k,h}(s_{k,h}, a_{k,h}) \leq \tilde{O}(\sigma(N_{k,h}(s_{k,h}, a_{k,h}))),$$

*where $\tilde{O}(\cdot)$ hides multiplicative logarithmic terms.*

The challenge here is that $\overline{V}_{k,h+1}(s_{k,h+1})$ is obtained by sampling from the posterior of $\tilde{Q}(s_{k,h+1}, \cdot)$ at action $\widehat{a}$ $(:= \arg\max_a \widehat{Q}_{k,h}(s_{k,h}, a) + \sigma(N_{k,h}(s_{k,h}, a)))$ and not $a_{k,h+1}$ $(:= \arg\max_a \widehat{Q}_{k,h}(s_{k,h}, a))$. To get around this difficulty, we show in Lemma C.2 that $\sigma(N_{k,h}(s_{k,h}, \widehat{a}))^2 < 2\sigma(N_{k,h}(s_{k,h}, a_{k,h}))^2 \log(1/\delta)$ with a non-zero probability. Finally, using a probability boosting argument (Lemma E.1) we prove Lemma 2. Combined with Lemma 1 to obtain a high probability optimism error bound.

**Lemma 3** (Optimism error). *In Algorithm 1, with probability $1 - 2\delta$, the following holds for all $k \in [K]$ and $h \in [H]$,*

$$V_h^*(s_{k,h}) - \tilde{Q}_{k,h}(s_{k,h}, a_{k,h}) \leq \tilde{O}(\sigma(N_{k,h}(s_{k,h}, a_{k,h}))),$$

*where $\tilde{O}(\cdot)$ hides multiplicative logarithmic terms.*

**(c) Bounding estimation error.** In Q-learning, an estimate of the next stage value function (here, $\overline{V}_{h+1}$) is used to compute the target in order to update the $Q$-value for the current stage (here, the posterior mean $\widehat{Q}_h$). As a result, the error in the posterior mean for stage $h$ depends on the error in the value function estimates for $h + 1$.

**Lemma 4** (Posterior mean estimation error). *With probability at least $1 - \delta$, for all $k, h, s, a \in [K] \times [H] \times \mathcal{S} \times \mathcal{A}$,*

$$\widehat{Q}_{k,h}(s, a) - Q_h^*(s, a) \leq \sqrt{\sigma(N_{k,h}(s, a))^2 \eta} + \alpha_n^0 H + \sum_{i=1}^n \alpha_n^i \left( \overline{V}_{k_i,h+1}(s_{k_i,h+1}) - V_{h+1}^*(s_{k_i,h+1}) \right),$$

*where $n = N_{k,h}(s, a)$, and $\eta = \log(SAKH/\delta)$. And, $\alpha_n^i = \alpha_i \Pi_{j=i+1}^n (1 - \alpha_j)$, $i > 0$, with $\alpha_n^0 = \Pi_{j=1}^n (1 - \alpha_j)$.*

Conceivably, we should be able to apply the above lemma inductively to obtain an estimation error bound (Lemma 5). Lemma 2 again plays a crucial role in the above recursive bound.

**Lemma 5** (Cumulative estimation error.). *With probability at least $1 - \delta$, the following holds for all $h \in [H]$,*

$$\sum_{k=1}^K \left( \tilde{Q}_{k,h}(s_{k,h}, a_{k,h}) - V_h^{\pi_k}(s_{k,h}) \right) \leq O\left( H^2 \sqrt{SAT} \log(JSAT/\delta) \right).$$

**(c) Putting it all together.** To obtain the final regret bound, we simply sum up the optimism error bound in Lemma 3 for $K$ episodes and add it to the cumulative estimation error bound above.

## 5 CONCLUSION

We presented a posterior sampling-based approach for incorporating exploration in Q-learning. Our PSQL algorithm is derived from an insightful Bayesian inference framework and shows promising empirical performance in preliminary experiments. (Detailed experimental setup and empirical results on additional environments are provided in Appendix A.) We proved a $\tilde{O}(H^2\sqrt{SAT})$ regret bound

in a tabular episodic RL setting that closely matches the known lower bound. Future directions include a theoretical analysis of the vanilla version of PSQL (called PSQL* in experiments) that uses a single sample from next stage posterior in the target computation. The vanilla version outperforms Algorithm 1 empirically but is significantly harder to analyze. Another avenue is tightening the $H$ dependence in the regret bound; Appendix F outlines a sketch for improving it by $\sqrt{H}$ although at the expense of making the algorithm more complex. Further refinements are potentially achievable using techniques from Li et al. (2021a); Zhang et al. (2020).

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

**The Use of Large Language Models**   Commonly availaible LLM tools were only used to help to improve english writing, grammar and typeset in Latex. LLMs were not used to generate any research ideas or analysis present in this work.

# A   EXPERIMENTS

In section 4, we proved that our Q-learning with posterior sampling algorithm  PSQL enjoys regret bounds comparable to its UCB-counterparts, e.g., Jin et al. (2018). In this section, we present empirical results that validate our theory and compare the empirical performance of the posterior sampling approach against several benchmark UCB-based and randomized algorithms for reinforcement learning.

For the empirical studies, we use the vanilla version of posterior sampling, which we denote as PSQL* (see Figure 1). In this vanilla version, the target computation at a step $h$ is the default $z = r_h + \max_{a'} \tilde{Q}_{h+1}(s_{h+1}, a')$. As discussed in Section 3.2, PSQL modified this target computation to make it slightly optimistic, to deal with the challenges in theoretical analysis. Later, we also compare the empirical performance of PSQL and PSQL*. While the modified target computation does slightly deteriorate the performance of PSQL, in our experiments, it still performs significantly better compared to the benchmark UCB-based approach Jin et al. (2018).

Specifically, we compare the posterior sampling approach to the following three alorithms.

- UCBQL Jin et al. (2018) (Hoeffding version): the seminal work which gave the first UCB based Q-learning regret analysis.
- RLSVI Russo (2019): a popular randomized algorithm that implicitly maintains posterior distributions on Value functions.
- Staged-RandQL Tiapkin et al. (2023): a recently proposed randomized Q-learning based algorithm that uses randomized learning rates to motivate exploration.

**Environment description.**   We report the empirical performance of RL algorithms on two tabular environments described below. In each environment, we report each algorithm's average performance over 10 randomly sampled instances.

- (One-dimensional "chain" MDP:) An instance of this MDP is defined by two parameters $p \in [0.7, 0.95]$ and $S \in \{7, 8, 9, \ldots, 14\}$. In a random instance, $p$, & $S$ are chosen randomly from the given ranges. The resultant MDP environment is a chain in which the agent starts at state 0 (the far-left state), and state $S$ (the right-most state) is the goal state. At any given step $h$ in an episode, the agent can take "left" or "right" action. The transitions are to the state in the direction of the action taken with probability $p$, and in the opposite direction with probability $1 - p$.
- (Two-dimensional "grid-world" MDP, similar to FrozenLake environment in the popular Gymnasium library:) A random instance of this MDP is defined by a $4 \times 4$ grid with a random number of "hole" states placed at on the grid uniformly at random that the agent must avoid or else the episode ends without any reward. The agent starts at the upper-left corner, and the goal state is the bottom-right corner of the grid. There is at least one feasible path from the starting state to the goal state that avoids all hole states. At any given time step, the agent can take the "left", "right", "bottom" and "up" actions. After an action is taken, the agent has $1/3$ probability to transit to the direction of the action taken, and $1/3$ probability each to transit to the two perpendicular directions.

In both the above environments, the goal state carries the reward of $(H - h)/H$, where $H$ is the duration of the episode and $h$ is the time index within the episode at which the goal state is reached. No other state has any reward. The duration of an episode is set at $H = 32$ for all experiments.

**Findings.**   We observed that the performance of all the algorithms is sensitive to constants in the exploration bonuses or in the posterior variances. These constants were tuned such that the respective algorithms performed the best in the two environments. We made the following parameter choices for the algorithmic simulations for a fair comparison:

- $\delta$ is fixed for all algorithms as $0.05$.
- In UCBQL Jin et al. (2018), the Q-function estimates are initialized as the maximum value of any state in the environment ($=: V_{\max}$). The exploration bonus for any $h, s, a$ with visit counts as $n$ is given by

$$\sqrt{c \frac{V_{\max}^2 \log(SAT/\delta)}{n}},$$

with $c = 0.01$

- In PSQL and PSQL*, the Q-function posterior means are initialized as $V_{\max}$ (same as UCBQL) and the standard deviation of the posterior for any $h, s, a$ with $n$ visits is given by

$$\sqrt{c \frac{V_{\max}^2}{\max\{1, n\}}},$$

with $c = 0.02$.

- In RLSVI Russo (2019), the per-reward perturbation is a mean zero Gaussian with standard deviation for any $h, s, a$ with $n$ visits is given by,

$$\sqrt{c \frac{V_{\max}^2 \log(SAT/\delta)}{n+1}},$$

with $c = 0.005$.

- In Staged-RandQL Tiapkin et al. (2023), for the initialization of the Q-function estimates, we use a tighter upper bound of $(H - h)/H$ at step $h$ available in our environment, instead of the default $H - h$ suggested in their paper. We use $n_0 = 1/S$ and $r_0 = 1$ as in their paper.

Our results are summarized in Figure 2 and 3. The error bars represent one standard deviation interval around the mean cumulative regret of an algorithm over 10 runs on randomly generated instances of the environment. We observe that the randomized/posterior sampling algorithms PSQL*, RLSVI,

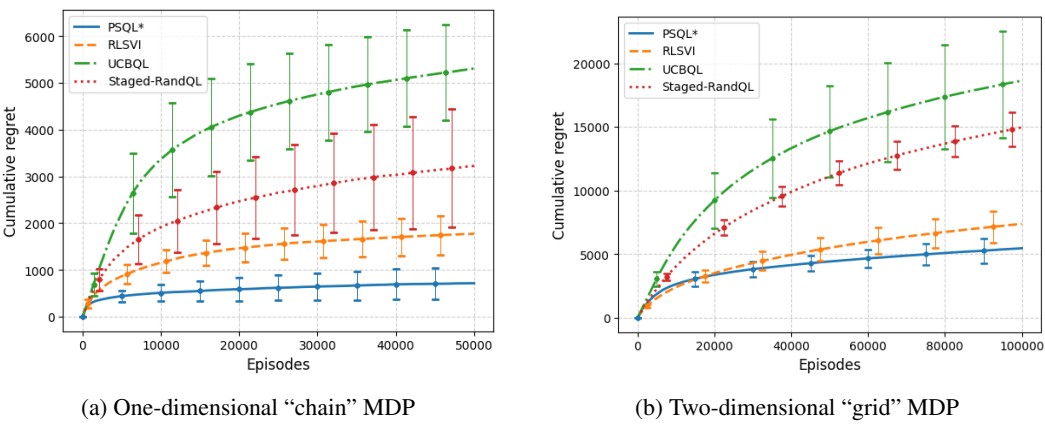

(a) One-dimensional "chain" MDP  (b) Two-dimensional "grid" MDP

Figure 2: Regret comparison: x-axes denotes episode index, y-axes denotes cumulative regret

and Staged-RandQL, have lower regret than their UCB counterpart: UCBQL. Also, PSQL* has significantly lower regret than the other two randomized algorithms.

A direct practical implication is that, PSQL* enjoys a shorter learning time (number of episodes after which the cumulative regret is below the specified threshold (Osband et al., 2019)). Further, the variance across different runs is also the lowest of all, suggesting PSQL* enjoys higher robustness.

In Figure 3, we compare the performance of PSQL*, the single sample vanilla version of posterior sampling, with the PSQL algorithm for which we provided regret bounds. As we explained in Section 4.1 (Challenges and Techniques), in order to achieve optimism in the target, PSQL computed the next state value by taking the max over multiple samples from the posterior of empirical mean maximizer

action $\widehat{a} = \arg\max_a \widehat{Q}_h(s_h, a) + \sigma(N_h(s_h, a))$. This introduces some extra exploration, and as a result, we observe that PSQL* displays a more efficient exploration-exploitation tradeoff, PSQL still performs significantly better than the UCB approach. These observations motivate an investigation into the theoretical analysis of the vanilla version, which we believe will require significantly new techniques.

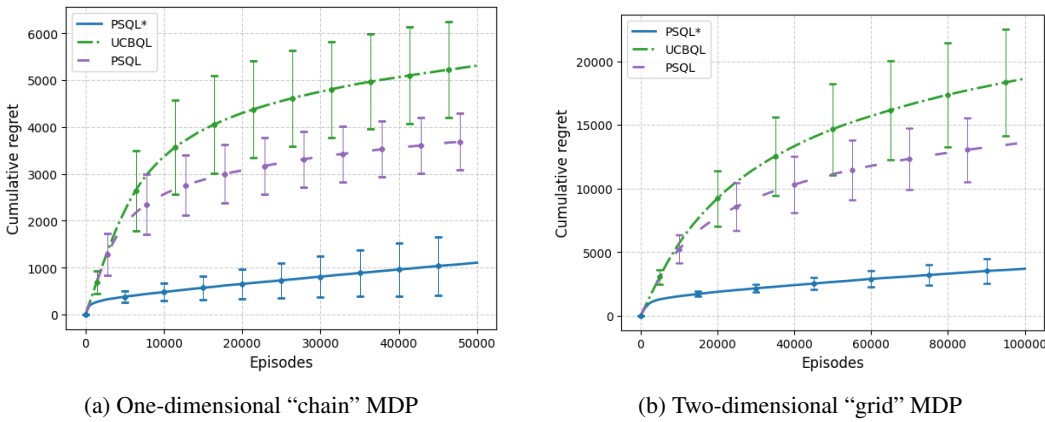

(a) One-dimensional "chain" MDP          (b) Two-dimensional "grid" MDP

Figure 3: Regret comparison: x-axes denotes episode index, y-axes denotes cumulative regret

## B    BAYESIAN INFERENCE BASED INTERPRETATION FOR Q-LEARNING

In this section, we describe the mathematical steps for calculating the updated posterior distribution from (8).

First, in Proposition B.1, we derive the well-known result that solving the optimization problem in (6) gives the posterior distribution as expected by Bayes rule. Let $\theta \in \Theta$ be the Bayesian parameter that we are inferring with $\Delta_\Theta$ be the space of distributions on $\Theta$. Let $p(\theta)$, $\ell(\theta, \cdot)$, and $q(\theta)$ be the the current prior distribution on $\theta$, the negative log likelihood function and the posterior distribution to be calculated.

**Proposition B.1** (Also in Khan & Rue (2023); Knoblauch et al. (2022)). *Let $KL(q(\theta)||p(\theta)) = \int_\theta q(\theta)\log(\frac{q(\theta)}{p(\theta)})$. Given log likellihood function $\ell(\theta, z)$, and prior $p(\theta)$, the distribution $q$ that maximizes ELBO objective,*

$$\max_{q \in \Delta_\Theta} \mathbb{E}_{\theta \sim q}[\ell(\theta, z)] - KL(q(\theta)||p(\theta)) \tag{14}$$

*is given by the Bayes rule*

$$q^{Bayes}(\theta) \propto p(\theta) \cdot \exp(\ell(\theta, z)). \tag{15}$$

*Proof.* Note that the ELBO objective function is equivalent to

$$-\int_\theta \log(\exp(-\ell(\theta, z)q(\theta) - \int_\theta \log\left(\frac{q(\theta)}{p(\theta)}\right)q(\theta)$$
$$= -\int_\theta \log\left(\frac{q(\theta)}{p(\theta)\exp(\ell(\theta, z))}\right)q(\theta),$$

which is maximized when $q(\theta) = q^{Bayes}(\theta)$.      $\square$

Now, we study the calculation of the posterior distribution of $Q_h(s, a)$ after observing $n + 1$ visits of $(h, s, a)$ in Lemma B.1.

**Lemma B.1.** *Consider the following maximization problem (regularized ELBO) over the space $\Delta_\Theta$ of distributions over a parameter $\theta$.*

$$\max_{q \in \Delta_\Theta} \mathbb{E}_{\theta \sim q}[\ell(\theta, z)] - KL(q(\theta)||p(\theta)) + \lambda_n \mathcal{H}(q(\theta)), \tag{16}$$

*Then, if $p(\cdot)$ is given by the pdf of the Gaussian distribution $\mathcal{N}(\widehat{\mu}_{n-1}, \frac{\sigma^2}{n-1})$, and $\ell(\theta, z) = \log(\phi_\theta(z))$ where $\phi_\theta(z) = \Pr(z|\theta)$ is the pdf of the Gaussian distribution $\mathcal{N}(\theta, \frac{\sigma^2}{H+1})$, and $\lambda_n = \frac{H}{n}$; then the optimal solution $q(\cdot)$ to (16) is given by the Gaussian distribution $\mathcal{N}(\widehat{\mu}_n, \frac{\sigma^2}{n})$, where*

$$\widehat{\mu}_n = (1 - \alpha_n)\widehat{\mu}_{n-1} + \alpha_n z, \ with \ \alpha_n = \frac{H+1}{H+n}.$$

*Proof.* Denote the objective value at a given distribution $q$ as $rELBO(q)$. Then,

$$
\begin{aligned}
rELBO(q) &= \int_\theta \log(\exp(\ell(\theta, z)q(\theta) - \int_\theta \log\left(\frac{q(\theta)}{p(\theta)}\right)q(\theta) - \lambda_n \int_\theta \log(q(\theta))q(\theta) \\
&= -\int_\theta \log\left(\frac{q(\theta)^{1+\lambda_n}}{p(\theta)\exp(\ell(\theta, z))}\right)q(\theta),
\end{aligned}
$$

which is maximized at distribution $q$ with $q(\theta) \propto (p(\theta)\exp(\ell(\theta, z)))^{1/(\lambda_n+1)}$ Then,

$$
\begin{aligned}
q(\theta) &\propto \exp\left[\frac{1}{\lambda_n + 1}\left(-\frac{(n-1)(\theta - \widehat{\mu}_{n-1})^2}{2\sigma^2} - \frac{(H+1)(z-\theta)^2}{2\sigma^2}\right)\right] \\
&\propto \exp\left[-\frac{n}{H+n}\left(\frac{\theta^2(H+n) - 2\theta((n-1)\widehat{\mu}_{n-1} + (H+1)z)}{2\sigma^2}\right)\right] \\
&\propto \exp\left[-n\left(\frac{\theta^2 - 2\theta\widehat{\mu}_n}{2\sigma^2}\right)\right] \\
&\propto \exp\left[-n\left(\frac{(\theta - \widehat{\mu}_n)^2}{2\sigma^2}\right)\right]
\end{aligned}
$$

where $\widehat{\mu}_n = \frac{n-1}{H+n}\widehat{\mu}_{n-1} + \frac{H+1}{H+n}z = (1 - \alpha_n)\widehat{\mu}_{n-1} + \alpha_n z$. $\square$

## C  Missing Proofs from Section 4

### C.1  Optimism

**Lemma C.1** (Unabridged version). *The samples from the posterior distributions and the mean of the posterior distributions as defined in Algorithm 1, 2 satisfy the following properties: for any episode $k \in [K]$ and index $h \in [H]$,*

*(a) (Posterior distribution mean) For any given $s, a$, with probability at least $1 - \frac{2(k-1)\delta}{KH} - \frac{\delta}{KH}$,*

$$\widehat{Q}_{k,h}(s, a) \geq Q_h^*(s, a) - \sqrt{\sigma(N_{k,h}(s,a))^2}. \tag{17}$$

*(b) (Posterior distribution sample) For any given $s, a$, with probability at least $p_1$ ($p_1 = \Phi(-1)$) conditioned on (17) being true,*

$$\tilde{Q}_{k,h}(s, a) \geq Q_h^*(s, a). \tag{18}$$

*(c) (In Algorithm 2) With probability at least $1 - \frac{2k\delta}{KH}$, the following holds for all episodes $k' \leq k$*

$$\overline{V}_{k,h}(s_{k,h}) \geq V_h^*(s_{k,h}). \tag{19}$$

*Here $\delta$ is a parameter of the algorithm used to define the number of samples $J$ used to compute the target $\overline{V}$.*

*Proof.* We prove the lemma statement via induction over $k, h$.

**Base case:** $k = 1, h \in [H]$. Note that $n_{1,h}(s,a) = 0$ for all $s, a, h \in \mathcal{S} \times \mathcal{A} \times [H]$. Therefore, $\widehat{Q}_{1,h}(s,a) = H$ for all $s, a, h$ ((17) is trivially true). As $Q_h^*(s,a) \leq H$ for all $s, a, h$, therefore $\tilde{Q}_{1,H}(s,a) \geq Q_H^*(s,a)$, with probability at least $1/2$ ($> p_1$), i.e. (18) is true. By the choice of $J$ and Lemma E.2, (19) also follows.

**Induction hypothesis:** Given $k > 1, 1 \leq h \leq H$, assume that the statements (a),(b), and (c) are true for $1 \leq k' \leq k-1, h' \in [H]$, and for $k' = k, h+1 \leq h' \leq H$ .

**Induction step:** For $k, h$, we show (17) holds with probability $1 - \frac{2(k-1)\delta}{KH} - \frac{\delta}{KH}$, (18) holds with probability at least $p_1 = \Phi(-1)$ in the event (17) holds , and finally (19) holds with probability $1 - \frac{2k\delta}{KH}$.

In case $n_{k,h}(s,a) = 0$, then $\widehat{Q}_{k,h}(s,a) = H$ and $b_{k,h}(s,a) > 0$ and therefore by the same reasoning as in the base case, the induction statement holds. For the rest of the proof we consider $n_{k,h}(s,a) > 0$.

Let $n = N_{k,h}(s,a)$, then Algorithm 1 implies,

$$\widehat{Q}_{k,h}(s,a) = \alpha_n^0 H + \sum_{i=1}^n \alpha_n^i \left( r_{k_i,h} + \overline{V}_{k_i,h+1}(s_{k_i,h+1}) \right), \tag{20}$$

To prove the induction step for (17), consider the following using (12) and Bellman optimality equation.

$$
\begin{aligned}
\widehat{Q}_{k,h}(s,a) - Q_h^*(s,a) &= \sum_{i=1}^n \alpha_n^i \left( r_{k_i,h} - r_h(s,a) + \overline{V}_{k_i,h+1}(s_{k_i,h+1}) - P_{h,s,a} V_h^* \right) \\
&= \sum_{i=1}^n \alpha_n^i \left( r_{k_i,h} - r_h(s,a) + V_{h+1}^*(s_{k_i,h+1}) - P_{h,s,a} V_h^* \right) \\
&\quad + \sum_{i=1}^n \alpha_n^i \left( \overline{V}_{k_i,h+1}(s_{k_i,h+1}) - V_{h+1}^*(s_{k_i,h+1}) \right)
\end{aligned}
$$

(using (19) from induction hypothesis for $h + 1 \leq H$, with probability $1 - \dfrac{2(k-1)\delta}{KH}$)

(note that this is trivially true when $h + 1 = H + 1$ since $\overline{V}_{k,H+1} = V_{H+1}^* = 0$)

$$\geq \sum_{i=1}^n \alpha_n^i \left( r_{k_i,h} - r_h(s,a) + V_{h+1}^*(s_{k_i,h+1}) - P_{h,s,a} V_h^* \right)$$

(using Corollary D.2 with probability $1 - \dfrac{\delta}{KH}$)

$$\geq -4 \sqrt{\frac{H^3 \log(KH/\delta)}{n_{k,h}(s,a) + 1}}$$

$$= -\sqrt{\sigma(N_{k,h}(s,a))^2}, \tag{21}$$

Therefore, with a union bound we have with probability $1 - \frac{2(k-1)\delta}{KH} - \frac{\delta}{KH}$,

$$\widehat{Q}_{k,h}(s,a) \geq Q_h^*(s,a) - \sqrt{\sigma(N_{k,h}(s,a))^2} \tag{22}$$

When (22) holds, then from the definition of cumulative density of Gaussian distribution we get,

$$\Pr\left( \tilde{Q}_{k,h}(s,a) \geq \widehat{Q}_{k,h}(s,a) + \sqrt{\sigma(N_{k,h}(s,a))^2} \right) \geq \Phi(-1).$$

Now, we show $\tilde{Q}_{k,h}(s,\widehat{a}) \geq Q_h^*(s,a^*) = V_h^*(s)$ with probability at least $\Phi(-1) - \delta - \delta/H$, where

$$\widehat{a} = \arg\max_{a \in \mathcal{A}} \widehat{Q}_{k,h}(s,a) + \sigma(N(s,a)), \text{ and } a^* = \arg\max_{a \in \mathcal{A}} Q_h^*(s,a).$$

By the definition of $\widehat{a}$ and properties of Gaussian distribution:

$$\underbrace{\tilde{Q}_{k,h}(s,\widehat{a}) \geq \widehat{Q}_{k,h}(s,\widehat{a}) + \sigma(N(s,\widehat{a}))}_{\text{with probability at least } \phi(-1)} \geq \underbrace{\widehat{Q}_{k,h}(s,a^*) + \sigma(N(s,a^*)) \geq Q^*_{k,h}(s,a^*)}_{\text{with probability at least } 1 - 2(k-1)\delta/KH - \delta/KH}.$$

Setting $J \geq \frac{\log(KH/\delta)}{\log(1/(1-p_1))}$ in Algorithm 1, we use Lemma E.2 to show that with probability $1 - \frac{\delta}{KH}$

$$\overline{V}_{k,h}(s_{k,h}) \geq V^*_h(s_{k,h}). \tag{23}$$

Finally we use a union bound to combine (22) and (23) to prove (19) holds with probability at least $1 - \frac{2k\delta}{KH}$.

$\square$

### C.2 ACTION MISMATCH BOUND

**Lemma C.2** (Bounding action mismatch). *For a given $k, h, s_{k,h}$, and let $\widehat{a} := \arg\max_a \widehat{Q}_{k,h}(s_{k,h}, a) + \sqrt{\sigma(N_{k,h}(s_{k,h}, a))^2}$,*

$$\sigma(N_{k,h}(s_{k,h}, \widehat{a}))^2 < 2\sigma(N_{k,h}(s_{k,h}, a_{k,h}))^2 \log(1/\delta), \quad \textit{with probability at least } p_2,$$

*where $p_2 = \Phi(-2) - (A)\delta$, and $\delta$ as defined in Theorem 2.*

*Proof.* Consider the following partition of $A$ actions:

$$\overline{\mathcal{A}} := \left\{ a : 2\sigma(N(s,a))^2 \log(1/\delta) > \sigma(N(s,\widehat{a}))^2 \right\},$$
$$\underline{\mathcal{A}} := \left\{ a : 2\sigma(N(s,a))^2 \log(1/\delta) \leq \sigma(N(s,\widehat{a}))^2 \right\}.$$

Clearly, $\widehat{a} \in \overline{\mathcal{A}}$. We prove that with probability at least $\Phi(-2) - A\delta$, we have $\tilde{a} \in \overline{\mathcal{A}}$, so that $\sigma(N(s,\widehat{a}))^2 < 2\sigma(N(s,a))^2 \log(1/\delta)$.

By definition of $\widehat{a}$, we have that for all $a$,

$$\widehat{Q}(s,a) + \sqrt{\sigma(N(s,a))^2} \leq \widehat{Q}(s,\widehat{a}) + \sqrt{\sigma(N(s,\widehat{a}))^2}.$$

Also, by construction of $\underline{\mathcal{A}}$, we have that for $\forall a \in \underline{\mathcal{A}}$,

$$\widehat{Q}(s,a) + \sqrt{\sigma(N(s,a))^2} + \sqrt{2\sigma(N(s,a))^2 \log(1/\delta)} \leq \widehat{Q}(s,\widehat{a}) + \sqrt{\sigma(N(s,\widehat{a}))^2} + \sqrt{\sigma(N(s,\widehat{a}))^2}$$

From Gaussian tail bounds (see Corollary D.1) we have for $\forall a \in \mathcal{A}$,

$$\Pr\left( \tilde{Q}(s,a) \leq \widehat{Q}(s,a) + \sqrt{\sigma(N(s,a))^2} + \sqrt{2\sigma(N(s,a))^2 \log(1/\delta)} \right) \geq 1 - A\delta$$

so that $\forall a \in \underline{\mathcal{A}}$,

$$\Pr\left( \tilde{Q}(s,a) \leq \widehat{Q}(s,\widehat{a}) + 2\sqrt{\sigma(N(s,\widehat{a}))^2} \right) \geq 1 - A\delta$$

Also for the Gaussian random variable $\tilde{Q}(s,\widehat{a})$, we have with probability at least $\Phi(-2)$,

$$\tilde{Q}(s,\widehat{a}) > \widehat{Q}(s,\widehat{a}) + 2\sqrt{\sigma(N(s,\widehat{a}))^2},$$

Using a union bound on the last two events,, we get

$$\tilde{Q}(s,\widehat{a}) > \max_{a \in \underline{\mathcal{A}}} \tilde{Q}(s,a), \quad \text{with probability at least } \Phi(-2) - A\delta. \tag{24}$$

Since $\widehat{a}$ is in $\overline{\mathcal{A}}$, in the above scenario, the action $\tilde{a}$ that maximizes $\tilde{Q}(s,\cdot)$ must be in $\overline{\mathcal{A}}$. Therefore, $\sigma(N(s,\widehat{a}))^2 < 2\sigma(N(s,a))^2 \log(1/\delta)$ with probability $\Phi(-2) - A\delta$.

$\square$

**Proposition C.1** (High probability action mismatch). *For a given $k, h, s_{k,h}$, let $\widehat{a} := \arg\max_a \widehat{Q}_{k,h}(s_{k,h}, a)$. Then,*

$$\sigma(N_{k,h}(s_{k,h}, \widehat{a})) < \sigma(N_{k,h}(s_{k,h}, a_{k,h}))\sqrt{2\log(1/\delta)} + \frac{1}{p_2}\mathbb{E}_{a_{k,h}}[\sigma(N_{k,h}(s_{k,h}, a_{k,h}))\sqrt{2\log(1/\delta)}].$$

*where $p_2 = \Phi(-2) - (A)\delta$, and $\delta$ as defined in Theorem 2. Here $\mathbb{E}_{a_{k,h}}[\cdot]$ denotes expectation over $a_{k,h}$ given $s_{k,h}$ and the history before round $k, h$.*

*Proof.* This follows by using previous lemma along with Lemma E.1 with $\tilde{X} = \sigma(N_{k,h}(s_{k,h}, a_{k,h}))\sqrt{2\log(1/\delta)}$, $X^* = \sigma(N_{k,h}(s_{k,h}, \widehat{a}))$, and $\underline{X} = 0$. $\qquad\square$

### C.3 TARGET ESTIMATION ERROR BOUND

The following lemma characterizes the target estimation error.

**Lemma C.3** (Target estimation error). *In Algorithm 1, with probability $1 - 2\delta$, the following holds for all $k \in [K]$ and $h \in [H]$,*

$$\overline{V}_{k,h}(s_{k,h}) - \tilde{Q}_{k,h}(s_{k,h}, a_{k,h})$$
$$\leq 4\sigma(N_{k,h}(s_{k,h}, a_{k,h}))\log(JKH/\delta) + \frac{4}{p_2}\mathbb{E}_{k,h}[\sigma(N_{k,h}(s_{k,h}, a_{k,h}))]\log(JKH/\delta)$$
$$=: \frac{1}{p_2}F(k, h, \delta),$$

*where $p_2 = \Phi(-2) - (A)\delta$ and $\delta$ as defined in Theorem 2, $J$ is defined in (11), and $\mathbb{E}_a[\cdot]$ denotes expectation over the randomness in the action taken at $k, h$ conditioned on all history at the start of the $h_{th}$ step in the episode $k$ (i.e., only randomness is that in the sampling from the posterior distribution).*

*Proof.* We have $\arg\widehat{a} = \max_a \widehat{Q}_{k,h}(s_{k,h}, a)$. For the remainder of the proof, we drop $k, h$ from the subscript and denote $a_{k,h}$ by $\tilde{a}$. From Algorithm 1, $\overline{V}(s)$ is the maximum of $J$ samples drawn from a Gaussian distribution with mean as $\widehat{Q}(s, \widehat{a})$ and standard deviation as $\sigma(N(s, \widehat{a}))^2$ Using Gaussian tail bounds ( Corollary D.1) along with a union bound over $J$ samples, for any $\delta \in (0, 1)$, with probability at least $1 - \delta$,

$$\overline{V}(s) \leq \widehat{Q}(s, \widehat{a}) + \sqrt{2\sigma(N(s, \widehat{a}))^2 \log(J/\delta)}$$
$$\leq \tilde{Q}(s, \widehat{a}) + \sqrt{2\sigma(N(s, \widehat{a}))^2 \log(J/\delta)} + \sqrt{2\sigma(N(s, \widehat{a}))^2 \log(1/\delta)}$$

where $\tilde{Q}(s, \widehat{a})$ is the sample corresponding to the $\widehat{a}$ action drawn by the algorithm at the $k, h$. Using a union bound to combine the statements, with probability at least $1 - 2\delta$ we have,

$$\overline{V}(s) \leq \tilde{Q}(s, \widehat{a}) + 2\sqrt{2\sigma(N(s, \widehat{a}))^2 \log(J/\delta)}$$
$$\leq \tilde{Q}(s, \tilde{a}) + 2\sqrt{2\sigma(N(s, \widehat{a}))^2 \log(J/\delta)}. \tag{25}$$

To complete the proof, we use Proposition C.1, and an union bound over all $k, h$ to have the following with probability at least $1 - 2\delta$

$$\overline{V}_{k,h}(s_{k,h}) - \tilde{Q}_{k,h}(s_{k,h}, a_{k,h})$$
$$\leq 4\sigma(N_{k,h}(s_{k,h}, a_{k,h}))\sqrt{\log(JKH/\delta)\log(KH/\delta)} + \frac{4}{p_2}\mathbb{E}_{k,h}[\sigma(N_{k,h}(s_{k,h}, a_{k,h}))]\sqrt{\log(JKH/\delta)\log(KH/\delta)}.$$

$$\square$$

## C.4 OPTIMISM ERROR BOUND

**Corollary C.1** (Optimism error bound). *In Algorithm 1, with probability $1 - 3\delta$, the following holds for any $k \in [K]$ and $h \in [H]$,*

$$V_h^*(s_{k,h}) - \tilde{Q}_{k,h}(s_{k,h}, a_{k,h})$$
$$\leq 4\sigma(N_{k,h}(s_{k,h}, a_{k,h})) \log(JKH/\delta) + \frac{4}{p_2} \mathbb{E}_{k,h}[\sigma(N_{k,h}(s_{k,h}, a_{k,h}))] \log(JKH/\delta)$$
$$=: \quad \frac{1}{p_2} F(k, h, \delta),$$

*where $p_2 = \Phi(-2) - (A)\delta$ and $\delta$ as defined in Theorem 2, $J$ is defined in (11), and $\mathbb{E}_a[\cdot]$ denotes expectation over the randomness in the action taken at $k, h$ conditioned on all history at the start of the $h_{th}$ step in the episode $k$ (i.e., only randomness is that in the sampling from the posterior distribution).*

*Proof.* From Lemma C.3, we have with probability at least $1 - 2\delta$,

$$\overline{V}_{k,h}(s_{k,h}) - \tilde{Q}_{k,h}(s_{k,h}, a_{k,h})$$
$$\leq 4\sigma(N_{k,h}(s_{k,h}, a_{k,h})) \log(JKH/\delta) + \frac{4}{p_2} \mathbb{E}_{k,h}[\sigma(N_{k,h}(s_{k,h}, a_{k,h}))] \log(JKH/\delta)$$
$$=: \quad \frac{1}{p_2} F(k, h, \delta).$$

Further, Lemma C.1 (c) gives with probability at least $1 - \delta$,

$$V_h^*(s_{k,h}) \leq \overline{V}_{k,h}(s_{k,h}).$$

We complete the proof via a union bound. $\qquad \square$

**Corollary C.2.** *With probability $1 - \delta$, the following holds for all $h \in [H]$,*

$$\sum_{k=1}^K V_h^*(s_{k,h}) - \tilde{Q}_{k,h}(s_{k,h}, a_{k,h}) \quad \leq \quad \mathrm{O}\left(\sqrt{H^2 SAT}\chi\right),$$

*where $\chi = \log(JSAT/\delta)$.*

*Proof.* From Corollary C.1, we have for all $k, h$ simultaneously, with probability $1 - 3\delta$,

$$\overline{V}_{k,h}(s_{k,h}) - \tilde{Q}_{k,h}(s_{k,h}, a_{k,h})$$
$$\leq 4\sigma(N_{k,h}(s_{k,h}, a_{k,h})) \log(JKH/\delta) + \frac{4}{p_2} \mathbb{E}_{k,h}[\sigma(N_{k,h}(s_{k,h}, a_{k,h}))] \log(JKH/\delta).$$

By combining the definition of the variance in (10) with Corollary D.4, the result follows easily. $\quad \square$

## C.5 POSTERIOR MEAN ESTIMATION ERROR BOUND

**Lemma 4** (Posterior mean estimation error). *With probability at least $1 - \delta$, for all $k, h, s, a \in [K] \times [H] \times \mathcal{S} \times \mathcal{A}$,*

$$\widehat{Q}_{k,h}(s, a) - Q_h^*(s, a) \leq \sqrt{\sigma(N_{k,h}(s, a))^2 \eta} + \alpha_n^0 H + \sum_{i=1}^n \alpha_n^i \left(\overline{V}_{k_i, h+1}(s_{k_i, h+1}) - V_{h+1}^*(s_{k_i, h+1})\right),$$

*where $n = N_{k,h}(s, a)$, and $\eta = \log(SAKH/\delta)$. And, $\alpha_n^i = \alpha_i \Pi_{j=i+1}^n (1 - \alpha_j)$, $i > 0$, with $\alpha_n^0 = \Pi_{j=1}^n (1 - \alpha_j)$.*

*Proof.* First consider a fixed $k, h, s, a$. From (12) and Bellman optimality equation, we have (assume $n = N_{k,h}(s,a) \geq 1$),

$$
\begin{aligned}
\widehat{Q}_{k,h}(s,a) - Q_h^*(s,a) &= \sum_{i=1}^{n} \alpha_n^i \left( r_{k_i,h} - r_h(s,a) + \overline{V}_{k_i,h+1}(s_{k_i,h+1}) - P_{h,s,a} V_h^* \right) \\
&= \sum_{i=1}^{n} \alpha_n^i \left( r_{k_i,h} - r_h(s,a) + V_{h+1}^*(s_{k_i,h+1}) - P_{h,s,a} V_h^* \right) \\
&\quad + \sum_{i=1}^{n} \alpha_n^i \left( \overline{V}_{k_i,h+1}(s_{k_i,h+1}) - V_{h+1}^*(s_{k_i,h+1}) \right) \\
&\quad \text{(Using Corollary D.1 with probability } 1 - \delta) \\
&\leq \sqrt{\sigma(N_{k,h}(s,a))^2 \log(1/\delta)} + \sum_{i=1}^{n} \alpha_n^i \left( \overline{V}_{k_i,h+1}(s_{k_i,h+1}) - V_{h+1}^*(s_{k_i,h+1}) \right).
\end{aligned}
$$
(26)

When $N_{k,h}(s,a) = 0$, then trivially $\widehat{Q}_{k,h}(s,a) - Q_h^*(s,a) \leq H = \alpha_n^0 H$, and for $N_{k,h}(s,a) > 0$, then $\alpha_n^0 = 0$. Combining these two cases and with a union bound over all $s, a, h, k$, we complete the proof. $\qquad \square$

## C.6 CUMULATIVE ESTIMATION ERROR BOUND

**Lemma 5** (Cumulative estimation error.). *With probability at least $1 - \delta$, the following holds for all $h \in [H]$,*

$$
\sum_{k=1}^{K} \left( \tilde{Q}_{k,h}(s_{k,h}, a_{k,h}) - V_h^{\pi_k}(s_{k,h}) \right) \leq O\left( H^2 \sqrt{SAT} \log(JSAT/\delta) \right).
$$

*Proof.* For the purpose of writing this proof, define $\phi_{k,h} := \tilde{Q}_{k,h}(s_{k,h}, a_{k,h}) - V_h^*(s_{k,h})$, $\delta_{k,h} := \tilde{Q}_{k,h}(s_{k,h}, a_{k,h}) - V_h^{\pi_k}(s_{k,h})$, and $\beta_{k,h} := \overline{V}_{k,h}(s_{k,h}) - V_h^*(s_{k,h})$. Clearly $\delta_{k,h} \geq \phi_{k,h}$. Further, $v(n_{k,h}) \leftarrow \sigma(N_{k,h}(s_{k,h}, a_{k,h}))^2$.

Now consider,

$$
\begin{aligned}
\tilde{Q}_{k,h}(s_{k,h}, a_{k,h}) - V_h^{\pi_k}(s_{k,h}) &\leq \tilde{V}_{k,h}(s_{k,h}) - Q_h^{\pi_k}(s_{k,h}, a_{k,h}) \\
&= \tilde{Q}_{k,h}(s_{k,h}, a_{k,h}) - Q_{k,h}^*(s_{k,h}, a_{k,h}) + Q_{k,h}^*(s_{k,h}, a_{k,h}) - Q_1^{\pi_k}(s_{k,1}, a_{k,1}) \\
&\quad \text{(from Lemma 4 with probability } 1 - \delta, \text{ with } n \leftarrow N_{k,h}(s_{k,h}, a_{k,h})) \\
&\leq \alpha_n^0 H + \sqrt{2v(n_{k,h})\eta} + \sum_{i=1}^{n} \alpha_n^i \beta_{k_i,h+1} + P_{s_{k,h}, a_{k,h}} \cdot (V_{h+1}^* - V_{h+1}^{\pi_k}) \\
&= \alpha_n^0 H + \sqrt{2v(n_{k,h})\eta} + \sum_{i=1}^{n} \alpha_n^i \beta_{k_i,h+1} - \phi_{k,h+1} + \delta_{k,h+1} \\
&\quad + P_{s_{k,h}, a_{k,h}} \cdot (V_{h+1}^* - V_{h+1}^{\pi_k}) - (V_{h+1}^*(s_{k,h+1}) - V_{h+1}^{\pi_k}(s_{k,h+1})) \\
&\quad \text{From Lemma C.3 with probability } 1 - 2\delta \\
&\leq \alpha_n^0 H + \sqrt{2v(n_{k,h})\eta} + \frac{1}{p_2} \sum_{i=1}^{n} \alpha_n^i F(k_i, h+1, \delta) + m_{k,h} \\
&\quad + \sum_{i=1}^{n} \alpha_n^i \phi_{k_i,h+1} - \phi_{k,h+1} + \delta_{k,h+1},
\end{aligned}
$$
(27)

where

$$
m_{k,h} := P_{s_{k,h}, a_{k,h}} \cdot (V_{h+1}^* - V_{h+1}^{\pi_k}) - (V_{h+1}^*(s_{k,h+1}) - V_{h+1}^{\pi_k}(s_{k,h+1})).
$$

Now, we club all episodes together to have,

$$
\sum_{k=1}^{K} \delta_{k,h} \leq \sum_{k=1}^{K} \alpha_n^0 H + \sum_{k=1}^{K} \sqrt{4v(n_{k,h})\eta} + \frac{1}{p_2} \sum_{k=1}^{K} \sum_{i=1}^{n} \alpha_n^i F(k_i, h+1, \delta/KH)) + \sum_{h=1}^{H} \sum_{k=1}^{K} m_{k,h}
$$

$$+ \sum_{k=1}^{K} \sum_{i=1}^{n} \alpha_n^i \phi_{k_i, h+1} - \sum_{k=1}^{K} \phi_{k, h+1} + \sum_{k=1}^{K} \delta_{k, h+1}.$$

From Lemma E.5 (c), it follows $(a_{k,h} = \{\phi_{k,h}, F(k, h, \delta/KH)\})$ $\sum_{k=1}^{K} \sum_{i=1}^{n} \alpha_n^i a_{k,h+1} \leq (1 + 1/H) \sum_{k=1}^{K} a_{k,h+1}$. Further from the initialization,

$$\sum_{k=1}^{K} \alpha_{n_{k,h}}^0 H \leq \sum_{k=1}^{K} \mathbb{I}\{n_{k,h} = 0\} H \leq SAH = HSA.$$

Therefore, we have,

$$\sum_{k=1}^{K} \delta_{k,h} \leq \sum_{k=1}^{K} SAH + \sum_{k=1}^{K} \left( \sqrt{4v(n_{k,h})\eta} + m_{k,h} \right) + (1 + 1/H) \frac{1}{p_2} \sum_{k=1}^{K} F(k, h+1, \delta/KH)) +$$
$$+ (1 + 1/H) \sum_{k=1}^{K} \phi_{k,h+1} - \sum_{k=1}^{K} \phi_{k,h+1} + \sum_{k=1}^{K} \delta_{k,h+1}.$$

Unrolling the above $H$ times to have with a union bound over $k, h \in [K] \times [H]$ with probability $1 - \delta$ ($\delta$ is scaled by $1/KH$ due to the union bound):

$$\sum_{k=1}^{K} \tilde{Q}_{k,1}(s_{k,1}, a_{k,1}) - V_1^{\pi_k}(s_{k,1}) \leq eSAH^2 + e \sum_{h=1}^{H} \sum_{k=1}^{K} \left( \sqrt{4v(n_{k,h})\eta} + m_{k,h} \right)$$
$$+ \frac{e}{p_2} \sum_{h=1}^{H} \sum_{k=1}^{K} F(k, h+1, \delta/KH)), \qquad (28)$$

where we have used $\delta_{k,h} \geq \phi_{k,h}$ and $\delta_{k,H+1} = 0$. Now, we analyze each term on the right hand side of (28) one by one. From Corollary D.3,

$$\sum_{h=1}^{H} \sum_{k=1}^{K} \sqrt{4v(n_{k,h})\eta} \leq O\left( \sqrt{H^4 SAT\eta} \right).$$

From Corollary D.4 with probability at least $1 - \delta$,

$$\frac{1}{p_2} \sum_{h=1}^{H} \sum_{k=1}^{K} \sum_{i=1}^{n} F(k_i, h+1, \delta/KH) \leq \frac{1 + H}{Hp_2} \sum_{h=1}^{H} \sum_{k=1}^{K} F(k, h+1, \delta/KH)$$
$$\leq O(1) \cdot \sum_{h=1}^{H} \sum_{k=1}^{K} \sqrt{v(n_{k,h})\chi \log(KH/\delta)} \qquad (29)$$
$$\leq O\left( \sqrt{H^4 SAT\chi} \right), \qquad (30)$$

where $\chi = \log(JSAT/\delta)$.

Finally, from Lemma D.1, we have with probability $1 - \delta$

$$\sum_{h=1}^{H} \sum_{k=1}^{K} m_{k,h} \leq O\left( \sqrt{H^4 T \log(KH/\delta)} \right).$$

Combining the above, we complete the proof. □

**Theorem 2.** *The cumulative regret of PSQL (Algorithm 1,2) in K episodes satisfies*

$$Reg(K) := \left( \sum_{k=1}^{K} V_1^*(s_{k,1}) - V_1^{\pi_k}(s_{k,1}) \right) \leq O\left( H^2 \sqrt{SAT}\chi \right),$$

*with probability at least $1 - \delta$, where $\chi = \log(JSAT/\delta)$ and $T = KH$.*

*Proof.* First we combine Corollary C.2 and Lemma 5 to get with probability at least $1 - \delta$ ($\chi = \log(JSAT/\delta)$),

$$\sum_{k=1}^{K} V_1^*(s_{k,1}) - V_1^{\pi_k}(s_{k,1}) \leq O\left(\sqrt{H^4 SAT\chi}\right).$$

Observing that the rewards are bound, there $|\sum_{k=1}^{K} \sum_{h=1}^{H} R_h(s_{k,h}, a_{k,h}) - \sum_{k=1}^{K} V_1^{\pi_k}(s_{k,1})| \leq O(H\sqrt{T \log(1/\delta)})$ with probability at least $1 - \delta$. This completes the proof of the theorem. $\qquad\square$

## D    CONCENTRATION RESULTS

**Corollary D.1.** *For a given $k, h, s, a \in [K] \times [H] \times \mathcal{S} \times \mathcal{A}$ (let $n = N_{k,h}(s,a)$), with probability $1 - \delta$, it holds,*

$$|\tilde{Q}_{k,h}(s,a) - \widehat{Q}_{k,h}(s,a)| \quad \leq \quad \sqrt{2\sigma(N_{k,h}(s_{k,h}, a_{k,h}))^2 \log(1/\delta)}. \tag{31}$$

*Proof.* The result directly follows form Lemma E.6. $\qquad\square$

**Corollary D.2.** *For some given $k, h, s, a \in [K] \times [H] \times \mathcal{S} \times \mathcal{A}$, the following holds with probability at least $1 - \delta$ (with $n = N_{k,h}(s,a)$),*

$$|\sum_{i=1}^{n} \alpha_n^i \left(r_{k_i,h} - R(s,a) + V_{h+1}^*(s_{k_i,h+1}) - P_{s,a}V_{h+1}^*\right)| \quad \leq \quad 4\sqrt{\frac{H^3 \log(1/\delta)}{n+1}}.$$

*Proof.* Let $k_i$ denote the index of the episode when $(s,a)$ was visited for the $i_{\text{th}}$ time at step $h$. Set $x_i = \alpha_n^i(r_{k_i,h} - R_h(s,a) + V(s_{i+1}) - P_{s_i,a_i} \cdot V)$ and consider filtration $\mathcal{F}_i$ as the $\sigma-$field generated by all random variables in the history set $\mathcal{H}_{k_i,h}$. $r_{k_i,h} - R_h(s,a) + V(s_{i+1}) - P_{s_i,a_i} \cdot (V) \leq H + 1$. Using the definition of the learning rate (Lemma E.5 (b)), we have $\sum_i^n x_i^2 \leq H(H+1)^2/n$. We apply Azuma-Hoeffding inequality (see Lemma E.3) combined with a union bound over all $(s,a,h) \in \mathcal{S} \times \mathcal{A} \times [H]$ and all possible values of $n \leq K$ to get the following with probability at least $1 - \delta$,

$$|\sum_{i=1}^{n} x_i| \quad \leq \quad 2\sqrt{\frac{2H^3 \log(1/\delta)}{n}}.$$

We complete the proof using the observation $\frac{1}{n+1} \geq \frac{1}{2n}$, $n \geq 1$. $\qquad\square$

**Lemma D.1.** *with probability at least $1 - \delta$, the following holds*

$$\sum_{k=1}^{K} \sum_{h=1}^{H} P_{s_{k,h}, a_{k,h}} \cdot (V_{h+1}^* - V_{h+1}^{\pi_k}) - (V_{h+1}^*(s_{k,h+1}) - V_{h+1}^{\pi_k}(s_{k,h+1})) \leq H^2 \sqrt{2T \log(1/\delta)}$$

*Proof.* For $x_{k,h} = P_{s_{k,h}, a_{k,h}} \cdot (V_{h+1}^* - V_{h+1}^{\pi_k}) - (V_{h+1}^*(s_{k,h+1}) - V_{h+1}^{\pi_k}(s_{k,h+1}))$ and filtration set $\mathcal{H}_{k,h}$ where $k$ is the episode index, $\{x_{k,h}, \mathcal{H}_{k,h}\}$ forms a martingale difference sequence with $|x_{k,h}| \leq H$. We complete the proof using Lemma E.3 and a union bound. $\qquad\square$

**Lemma D.2.** *Let $\mathcal{D}_{k,h}$ be the distribution of actions at time step $k, h$ conditioned on the history at the start of step $h$ of the $k_{th}$ episode, then with probability at least $1 - \delta$, for some $h$*

$$\sum_{k=1}^{K} \mathbb{E}_{a \sim \mathcal{D}_{k,h}} \left[\frac{1}{\sqrt{n_{k,h}(s_{k,h}, a) + 1}}\right] - \frac{1}{\sqrt{n_{k,h}(s_{k,h}, a_{k,h}) + 1}} \leq O\left(\sqrt{SA \log(K)} \log(1/\delta)\right)$$

*Proof.* For brevity of exposition, let $\mathbb{E}[Z_k] \leftarrow \mathbb{E}_{a \sim \mathcal{D}_{k,h}}\left[\frac{1}{\sqrt{n_{k,h}(s_{k,h},a)+1}}\right]$, $Z_k \leftarrow \frac{1}{\sqrt{n_{k,h}(s_{k,h},a_{k,h})+1}}$, and $\mathcal{F}_k \leftarrow \mathcal{H}_{k,h}$. Consider,

$$
\begin{aligned}
\sum_{k=1}^{K}(\mathbb{E}[Z_k] - Z_k)^2 &\leq (\mathbb{E}[Z_k])^2 + Z_k^2 \\
&\qquad \text{(By Jensen's inequality for } f(x) = x^2) \\
&\leq \sum_{k=1}^{K} Z_k^2 + \sum_{k=1}^{K} \mathbb{E}[Z_k^2] \\
&\qquad \text{(by linearity of expectation)} \\
&= \sum_{k=1}^{K} Z_k^2 + \mathbb{E}\left[\sum_{k=1}^{K} Z_k^2\right] \\
&\leq \sum_{s,a}\sum_{j=1}^{K} \frac{1}{j+1} + \mathbb{E}\left[\sum_{s,a}\sum_{j=1}^{K} \frac{1}{j+1}\right] \\
&\leq 2SA\log(K).
\end{aligned}
$$

To bound $\sum_{k=1}^{K}(\mathbb{E}[Z_k] - Z_k)$, we apply Bernstein inequality for martingale, Lemma E.4, with $K = 1$, $d = 2SA\log(K)$ to get the required result. $\qquad\square$

**Lemma D.3.**

$$
\sum_{k=1}^{K} \frac{1}{\sqrt{N_{k,h}(s_{k,h}, a_{k,h}) + 1}} \leq \mathrm{O}\left(\sqrt{SAK}\right)
$$

*Proof.*

$$
\begin{aligned}
\sum_{k=1}^{K} \frac{1}{\sqrt{N_{k,h}(s_{k,h}, a_{k,h}) + 1}} &\leq \sum_{k=1}^{K} \frac{\sqrt{2}}{\sqrt{N_{k,h}(s_{k,h}, a_{k,h})}} \\
&\leq \mathrm{O}(1)\sum_{s,a}\sum_{k=1}^{N_{K,h}(s,a)} \sqrt{\frac{1}{k}} \\
&\leq \mathrm{O}\left(\sqrt{SAK}\right).
\end{aligned}
$$

$\qquad\square$

**Corollary D.3.** *The following holds,*

$$
\sum_{h=1}^{H}\sum_{k=1}^{K} \sqrt{4\sigma(N_{k,h}(s_{k,h}, a_{k,h}))^2 \eta} \leq \mathrm{O}\left(\sqrt{H^4 SAT\eta}\right).
$$

*Proof.* From Lemma D.3 and (10), we get the result. $\qquad\square$

**Corollary D.4.** *Let $\mathcal{D}_{k,h}$ be the distribution of actions at time step $k, h$ conditioned on the history at the start of step $h$ of the $k_{th}$ episode, then with probability at least $1 - \delta$*

$$
\sum_{k=1}^{K} \frac{4}{p_2} \mathbb{E}_{a \sim \mathcal{D}_{k,h}}\left[\sqrt{\sigma(N_{k,h}(s_{k,h}, a))^2 \log(JKH/\delta)}\right] + \sqrt{2\sigma(N_{k,h}(s_{k,h}, a_{k,h}))^2 \log(JKH/\delta)}
$$

$$
\leq \quad O\left(\sqrt{H^2 SAT\chi}\right),
$$

$\chi = \log(JSAT/\delta)$.

*Proof.* Using Lemma D.2, we have with probability $1 - \delta$,

$$\sum_{k=1}^{K} \mathbb{E}_{a \sim \mathcal{D}_{k,h}} \left[ \frac{1}{\sqrt{n_{k,h}(s_{k,h}, a) + 1}} \right] - \frac{1}{\sqrt{n_{k,h}(s_{k,h}, a_{k,h}) + 1}} \le O\left( \sqrt{SA \log(K)} \log(1/\delta) \right)$$

From Lemma D.3 and (10),

$$\sum_{k=1}^{K} \frac{4}{p_2} \sqrt{\sigma(N_{k,h}(s_{k,h}, a))^2} \log(JKH/\delta) \le O\left( \sqrt{H^2 SAT\chi} \right),$$

which dominates the remaining terms. $\qquad \square$

## E TECHNICAL PRELIMINARIES

**Lemma E.1** (High confidence from constant probability). *For some fixed scalars $X^*, \underline{X}$, and $p, \delta \in (0, 1)$, suppose that $\tilde{X} \sim \mathcal{D}$ satisfies $\tilde{X} \ge X^*$ with probability at least $p$, $\tilde{X} \ge \underline{X}$ with probability at least $1 - \delta$, and $\mathbb{E}[\tilde{X}] \ge \underline{X}$. Then, with probability at least $1 - 2\delta$,*

$$\tilde{X} \ge X^* - \tfrac{1}{p} \left( \mathbb{E}_{\mathcal{D}}[\tilde{X}] - \underline{X} \right). \tag{32}$$

*Proof.* For the purpose of this proof, a symmetric sample $\tilde{X}^{\text{alt}}$ also drawn from distribution $\mathcal{D}$ but independent of $\tilde{X}$. Let $\mathcal{O}^{\text{alt}}$ denotes the event when $\tilde{X}^{\text{alt}} \ge X^*$ (occurring with probability $p$).

Consider (using notation $\mathbb{E}[\cdot] \leftarrow \mathbb{E}_{\mathcal{D}}[\cdot]$)

$$X^* - \tilde{X} \le \; \mathbb{E}\left[ \tilde{X}^{\text{alt}} \mid \mathcal{O}^{\text{alt}} \right] - \tilde{X} \le \; \mathbb{E}\left[ \tilde{X}^{\text{alt}} - \underline{X} \mid \mathcal{O}^{\text{alt}} \right], \tag{33}$$

where the last inequality holds with probability $1 - \delta$ by definition of $\underline{X}$. The law of total expectation suggests,

$$\mathbb{E}\left[ \tilde{X}^{\text{alt}} - \underline{X} \right] \; = \; \Pr(\mathcal{O}^{\text{alt}}) \mathbb{E}\left[ \tilde{X}^{\text{alt}} - \underline{X} \mid \mathcal{O}^{\text{alt}} \right] + \Pr(\overline{\mathcal{O}}^{\text{alt}}) \mathbb{E}\left[ \tilde{X}^{\text{alt}} - \underline{X} \mid \overline{\mathcal{O}}^{\text{alt}} \right],$$

where $\overline{\mathcal{O}}^{\text{alt}}$ is the compliment of the event $\mathcal{O}^{\text{alt}}$. Now, $\mathbb{E}\left[ \tilde{X}^{\text{alt}} \mid \overline{\mathcal{O}}^{\text{alt}} \right] = \mathbb{E}\left[ \tilde{X}^{\text{alt}} \mid \tilde{X}^{\text{alt}} < X^* \right] \le \mathbb{E}[\tilde{X}^{\text{alt}}] = \mathbb{E}[\tilde{X}] \ge \underline{X}$, where the last inequality is by the assumption made in the lemma. Therefore, the second term in the above is non-negative, and we have

$$\mathbb{E}\left[ \tilde{X}^{\text{alt}} - \underline{X} \right] \; \ge \; \Pr(\mathcal{O}^{\text{alt}}) \mathbb{E}\left[ \tilde{X}^{\text{alt}} - \underline{X} \mid \mathcal{O}^{\text{alt}} \right]. \tag{34}$$

Using $\frac{1}{\Pr(\mathcal{O}^{\text{alt}})} \le \frac{1}{p}$, $\mathbb{E}[\tilde{X}^{\text{alt}}] = \mathbb{E}[\tilde{X}]$, and a union bound to combine (33) and (34), we complete the proof. $\qquad \square$

**Lemma E.2.** *Let $q^{(1)}, q^{(2)}, \ldots q^{(M)}$ be $M$ i.i.d. samples such that for any $i$, $q^{(i)} \ge V^*$ with probability $p$. Then with probability at least $1 - \delta$,*

$$\max_{i \in M} q^{(i)} \ge V^*,$$

*when $M$ is at least $\frac{\log(1/\delta)}{\log(1/(1-p))}$.*

*Proof.* For a given index $i$, the probability that $q^{(i)} < V^*$ is at most $1 - p$. Therefore, by independence of samples, the probability of $\max_{i \in J} q^{(i)} < V^*$ is at most $(1 - p)^M$. Therefore, the lemma statement follows by setting $M = \frac{\log(1/\delta)}{\log(1/(1-p))}$. $\qquad \square$

**Lemma E.3** (Corollary 2.1 in Wainwright (2019)). *Let $(\{A_i, \mathcal{F}_i\}_{i=1}^{\infty})$ be a martingale difference sequence, and suppose $|A_i| \le d_i$ almost surely for all $i \ge 1$. Then for all $\eta \ge 0$,*

$$\mathbb{P}\left[ |\sum_{i=1}^{n} A_i| \ge \eta \right] \le 2 \exp\left( \frac{-2\eta^2}{\sum_{i=1}^{n} d_i^2} \right). \tag{35}$$

*In other words, with probability at most $\delta$, we have,*

$$|\sum_{i=1}^{n} A_i| \geq \sqrt{\frac{ln\left(2/\delta\right)\sum_{i=1}^{n} d_i^2}{2}} \tag{36}$$

**Lemma E.4** (Lemma A8 in Cesa-Bianchi & Lugosi (2006)). *Let $(\{A_i, \mathcal{F}_i\}_{i=1}^{\infty})$ be a martingale difference sequence, and suppose $|A_i| \leq K$ almost surely for all $i \geq 1$. Let $S_i = \sum_{j=1}^{i} A_i$ be the associated martingale. Denote the sum of the conditional variances by*

$$v_n^2 = \sum_{t=1}^{n} \mathbb{E}[A_i^2 \mid \mathcal{F}_{1-1}].$$

*Then for all constants $t, d > 0$,*

$$\mathbb{P}\left[\max_i S_i \geq t \,\&\,, v_n^2 \leq d\right] \leq \exp\left(-\frac{t^2}{2(d + Kt/3)}\right).$$

**Lemma E.5** (Lemma 4.1 in Jin et al. (2018)). *The following holds:*

(a) $\frac{1}{\sqrt{n}} \leq \sum_{i=1}^{n} \frac{\alpha_n^i}{\sqrt{i}} \leq \frac{2}{\sqrt{n}}$.

(b) $\max_{i \in n} \alpha_n^i \leq \frac{2C}{t}$ *and* $\sum_{i=1}^{n} (\alpha_n^i)^2 \leq \frac{2C}{t}$.

(c) $\sum_{n=i}^{\infty} \alpha_n^i \leq 1 + 1/C$.

**Lemma E.6** (Gaussian tail bound). *For a Gaussian random variable $X \sim \mathcal{N}(\mu, \sigma^2)$, it follows with probability at least $1 - \delta$,*

$$Pr\left(|X - \mu| \leq \sigma\sqrt{2\log(1/\delta)}\right)$$

*Proof.* The proof follows by instantiating Chernoff-style bounds for the given Gaussian random variable. □

## F SHARPER REGRET USING BERNSTEIN CONCENTRATION

In this section, we provide a sketch of the extension of Algorithm 1 to a randomized Q-learning procedure that uses Bernstein concentration based variance. This extension closely follows that in Jin et al. (2018) using some of the techniques developed in the proof of Theorem 2.

We want to account for the variance in the transitions. To this end, we define some additional notations. The variance in transition for any $(s, a)$ is defined using the variance operator $\mathbb{V}_h$ as below,

$$[\mathbb{V}_h V_{h+1}]_{s,a} := \mathbb{E}_{s' \sim P_{h,s,a}}\left[V_{h+1}(s') - [P_{h,s,a} V_{h+1}]_{s,a}\right]^2. \tag{37}$$

The empirical variance for any $(s, a)$ for any $n \leftarrow N_{k,h}(s, a)$ is given by,

$$\widehat{\mathbb{V}}_n \overline{V}_{h+1}(s,a) = \frac{1}{n}\sum_{i=1}^{n}\left[\overline{V}_{k_i,h+1}(s_{k_i,h+1}) - \frac{1}{n}\sum_{i=1}^{n}\overline{V}_{k_i,h+1}(s_{k_i,h+1})\right]^2 \tag{38}$$

$$\sqrt{v_b(n,h,s,a)} := \min\left\{c(\sqrt{\frac{H}{n+1}}\cdot(\widehat{\mathbb{V}}_n\overline{V}_{h+1}(s,a) + H)\eta + \frac{\sqrt{H^7 SA\eta}\cdot\tau}{n+1}), \sqrt{64\frac{H^3}{n+1}}\right\}. \tag{39}$$

**Theorem F.1.** *The cumulative regret of Algorithm 3 in $K$ episode satisfies with probability at least $1 - \delta$,*

$$\sum_{k=1}^{K} V_1^*(s_{k,1}) - \sum_{k=1}^{K}\sum_{h=1}^{H} R_h(s_{k,h}, a_{k,h}) \leq O\left(\sqrt{H^3 SAT\eta\chi}\right),$$

*where $\chi = \log(JSAT/\delta)$ and $\eta = \log(SAKH/\delta)$.*

---

**Algorithm 3** Randomized Q-learning

1: **Input:** Parameters: $\delta \in (0,1)$. Set $J := J(\delta)$.
2: **Initialize:** $\widehat{Q}_{H+1}(s,a) = \widehat{V}_{H+1}(s) = 0, \forall (s,a) \in \mathcal{S} \times \mathcal{A}$, and $\widehat{Q}_h(s,a) = \widehat{V}_h(s) = H$ & $N_h(s,a) = 0, N_h^{\text{tar}}(s) = 0, \mu_h(s,a) = \gamma_h(s,a) = 0 \,\forall (s,a,h) \in \mathcal{S} \times \mathcal{A} \times [H]$.

3: **for** episodes $k = 1, 2, \ldots$ **do**
4:     Observe $s_1$.
5:     **for** step $h = 1, 2, \ldots, H$ **do**

6:         /* Play arg max action of the sample of $Q_h$ */
7:         Sample $\forall a \in \mathcal{A}\, \tilde{Q}_h(s_h,a) \sim \mathcal{N}(\widehat{Q}_h(s_h,a), v_b(N_h(s_h,a),h,s_h,a))$.
8:         Play $a_h = \arg\max_{a_{\mathcal{A}}} \tilde{Q}_h(s_h,a)$.

9:         Use observations to construct one step lookahead target $z$ */
10:        Observe $r_h$ and $s_{h+1}$.
11:        $z \leftarrow \texttt{ConstructTarget}(r_h, s_{h+1}, \widehat{Q}_{H+1}, N_{H+1})$.

12:        /*Use the observed reward and next state to update $Q_h$ distribution */
13:        $n := N_h(s_h,a_h) \leftarrow N_h(s_h,a_h) + 1$.
14:        $\widehat{Q}_h(s_h,a_h) \leftarrow (1-\alpha_n)\widehat{Q}_h(s_h,a_h) + \alpha_n z$.
15:        $\mu_h(s_h,a_h) \leftarrow \mu_h(s_h,a_h) + (z - r_h)$.
16:        $\gamma_h(s_h,a_h) \leftarrow \gamma_h(s_h,a_h) + (z - r_h)^2$.
17:        Calculate $b_{h+1}(s_h,a_h) \leftarrow (\gamma_h(s_h,a_h) - \mu_h(s_h,a_h)^2)/n$.
18:        $\sqrt{v_b(n,h,s_h,a_h)} \leftarrow \min\{c(\sqrt{\frac{H}{n+1} \cdot (b_{h+1}(s_h,a_h)+H)\eta} + \frac{\sqrt{H^7 SA\eta\chi}}{n+1}), \sqrt{64\frac{H^3}{n+1}}\}$.
19:     **end for**
20: **end for**

---

### F.1 PROOF OF THEOREM F.1

The main mathematical reasoning closely follows that in the proof of Theorem 2 of Jin et al. (2018) with specific differences arising due to constant probability optimism and the definition of $\overline{V}_{k,h}$. For any $k, h, s, a$ with $n = N_{k,h}(s,a)$, we have $v_b(n,h,s,a) \le 64\frac{H^3}{n+1}$. Therefore, Corollary C.2 and Lemma 5 apply as they are. Hence, we get with probability at least $1 - \delta$ for all $h$

$$\sum_{k=1}^{K} r_{k,h} := \sum_{k=1}^{K} V_h^*(s_{k,h}) - V_h^{\pi_k}(s_{k,h}) \le O\left(\sqrt{H^4 SAT\chi}\right), \tag{40}$$

where $\chi = \log(JSAT/\delta)$. Further, following the steps and notations of the proof of Lemma 5 (see (28), we have with probability at least $1 - \delta$,

$$\sum_{k=1}^{K} \tilde{Q}_{k,1}(s_{k,1}, a_{k,1}) - V_1^{\pi_k}(s_{k,1}) \le eSAH^2 + e\sum_{h=1}^{H}\sum_{k=1}^{K}\left(\sqrt{4v_b(n,h,s_{k,h},a_{k,h})\eta} + m_{k,h}\right)$$

$$+ \frac{e}{p_2}\sum_{h=1}^{H}\sum_{k=1}^{K} F(k, h+1, \delta/KH)). \tag{41}$$

Due to our observation that $v_b(n,h,s_{k,h},a_{k,h}) \le 64\frac{H^3}{n+1}$, Lemma D.2 and Corollary D.4 hold, therefore we get from (41),

$$\sum_{k=1}^{K} \tilde{Q}_{k,1}(s_{k,1}, a_{k,1}) - V_1^{\pi_k}(s_{k,1}) \le eSAH^2 + \sum_{h=1}^{H}\sum_{k=1}^{K} O\left(\sqrt{v_b(n,h,s_{k,h},a_{k,h})\chi} + m_{k,h}\right),$$

where $\chi = \log(JSAT/\delta)$. We wish to bound

$$\sum_{k=1}^{K}\sum_{h=1}^{H} v_b(n,h,s_{k,h},a_{k,h}) \tag{42}$$

$$\le \sum_{k=1}^{K}\sum_{h=1}^{H} c(\sqrt{\frac{H}{n+1} \cdot (\widehat{\mathbb{V}}_n\overline{V}_{h+1}(s_{k,h},a_{k,h})+H)\eta} + \frac{\sqrt{H^7 SA\eta}\cdot\tau}{n+1}), \tag{43}$$

Consider,

$$\sum_{k=1}^{K}\sum_{h=1}^{H}\frac{\sqrt{H^7 SA\eta\chi}}{N_{k,h}(s_{k,h},a_{k,h})+1} \leq O(\sqrt{H^9 S^3 A^3 \chi^5}). \tag{44}$$

Further,

$$\sum_{k=1}^{K}\sum_{h=1}^{H}\sqrt{\frac{H}{n+1}\cdot(\widehat{\mathbb{V}}_n \overline{V}_{k,h+1}(s_{k,h},a_{k,h})+H)}$$

$$\leq \quad \sum_{k=1}^{K}\sum_{h=1}^{H}\sqrt{\frac{H}{n+1}\cdot\widehat{\mathbb{V}}_n \overline{V}_{k,h+1}(s_{k,h},a_{k,h})} + \sum_{k=1}^{K}\sum_{h=1}^{H}\sqrt{\frac{H^2}{n+1}}$$

$$\leq \quad \sqrt{\sum_{k=1}^{K}\sum_{h=1}^{H}\widehat{\mathbb{V}}_n \overline{V}_{k,h+1}(s_{k,h},a_{k,h})H} + \sqrt{H^3 SAT\eta}, \tag{45}$$

where the last inequality follows from Lemma D.3. Since we have Lemma F.5, we can follow the steps in (C.16) of Jin et al. (2018) to get

$$\sum_{k=1}^{K}\sum_{h=1}^{H}\widehat{\mathbb{V}}_n \overline{V}_{k,h+1}(s_{k,h},a_{k,h}) \leq O(HT).$$

This gives us

$$\sum_{k=1}^{K}\sum_{h=1}^{H}v_b(n,h,s_{k,h},a_{k,h}) \leq O(\sqrt{H^3 SAT\eta} + \sqrt{H^9 S^3 A^3 \chi^5})$$

Thus we have,

$$\sum_{k=1}^{K}\tilde{Q}_{k,1}(s_{k,1},a_{k,1}) - V_1^{\pi_k}(s_{k,1}) \leq O(\sqrt{H^3 SAT\eta}\chi + \sqrt{H^9 S^3 A^3 \chi^5})$$

Finally, combining with Corollary C.2, we complete the proof.

### F.2 SUPPORTING LEMMA

**Corollary F.1** (Corollary of Lemma 4). *We have for all $s,a,h,k \in \mathcal{S}\times\mathcal{A}\times[H]\times[K]$ with probability at least $1-\delta$,*

$$\widehat{Q}_{k,h}(s,a) - Q_h^*(s,a) \leq O\left(\sqrt{\frac{H^3\eta}{n+1}}\right) + \alpha_n^0 H + \sum_{i=1}^{n}\alpha_n^i\left(\overline{V}_{k_i,h+1}(s_{k_i,h+1}) - V_{h+1}^*(s_{k_i,h+1})\right), \tag{46}$$

*where $n = N_{k,h}(s,a)$, and $\eta = \log(SAKH/\delta)$.*

*Proof.* The proof follows from using (39) in place of $\sigma(N_{k,h}(s,a))^2$ in Lemma 4. $\square$

**Lemma F.1** (Based on Lemma C.7 in Jin et al. (2018)). *Suppose (46) in Corollary F.1 holds. For any $h \in [H]$, let $\beta_{k,h} := \overline{V}_{k,h}(s_{k,h}) - V_h^*(s_{k,h})$ and let $w = (w_1,\ldots,w_k)$ be non-negative weight vectors, then we have with probability at least $1-\delta$,*

$$\sum_{k=1}^{K}w_k\beta_{k,h} \leq O(SA||w||_\infty\sqrt{H^7}\chi^2 + \sqrt{SA||w||_1||w||_\infty H^5}\chi),$$

*where $\chi = \log(JSAT/\delta)$.*

*Proof.* For any fixed $k, h$, let $n \leftarrow N_{k,h}(s_{k,h}, a_{k,h})$ and $\phi_{k,h} = \tilde{Q}_{k,h}(s_{k,h}, a_{k,h}) - V_h^*(s_{k,h})$. Then we have

$$
\begin{aligned}
\beta_{k,h} &= \overline{V}_{k,h}(s_{k,h}) - V_h^*(s_{k,h}) \\
&\quad \text{(from Lemma C.3 with probability at least } 1 - 2\delta) \\
&\leq \phi_{k,h} + \frac{1}{p2}F(k,h,\delta) \\
&\quad \text{(from Lemma C.5 with probability } 1 - \delta) \\
&\leq \widehat{Q}_{k,h}(s_{k,h}, a_{k,h}) - Q_h^*(s_{k,h}, a_{k,h}) + \sqrt{2v(n)\eta} + \frac{1}{p_2}F(k,h,\delta) \\
&\quad \text{(from Corollary F.1 with probability } 1 - \delta) \\
&\leq \alpha_n^0 H + O\left(\sqrt{\frac{H^3\eta}{n+1}}\right) + \sum_{i=1}^n \alpha_n^i \beta_{k_i,h+1} + \frac{1}{p_2}F(k,h,\delta) \quad (47)
\end{aligned}
$$

We now compute the summation $\sum_{k=1}^K w_k \beta_{k,h}$. We follow the proof of Lemma C.7 in Jin et al. (2018), with the only difference being the term $\sum_{k=1}^K \frac{w_k}{p_2}F(k,h,\delta)$, which we bound below. From the proof of Corollary D.4.

$$
\begin{aligned}
\sum_{k=1}^K \frac{w_k}{p_2}F(k,h,\delta) &\leq \frac{\|w\|_\infty}{p_2}O(SA\sqrt{H^5\log^4(JSAK/\delta)}) + \sum_{k=1}^K \frac{2w_k}{p_2}\left(\sqrt{\frac{H^3\chi\log(KH/\delta)}{N_{k,h}+1}}\right) \\
&\leq O(SA\|w\|_\infty\sqrt{H^5\chi^4} + \sqrt{SA\|w\|_1\|w\|_\infty H^3}\chi),
\end{aligned}
$$

where $\chi = \log(JSAT/\delta)$. Other terms are evaluated in the same way as the proof of Lemma C.7 in Jin et al. (2018). $\quad\square$

**Lemma F.2** (Based on Lemma C.3 of Jin et al. (2018))**.** *For any episode* $k \in [K]$ *with probability* $1 - \delta/K$, *if Corollary F.1 holds for all* $k' < k$, *the for all* $s, a, h \in \mathcal{S} \times \mathcal{A} \times [H]$:

$$
\left|[\mathbb{V}_h V_{h+1}]_{s,a} - \widehat{\mathbb{V}}_n \overline{V}_{h+1}(s,a)\right| \leq O\left(\frac{SA\sqrt{H^9}\chi^2}{n} + \sqrt{\frac{H^7SA\chi^2}{n}}\right),
$$

*where* $n = N_{k,h}(s_{k,h}, a_{k,h})$, $\chi = \log(JSAT/\delta)$.

*Proof.* The proof is almost identical to that of Lemma C.3 of Jin et al. (2018) except we Lemma F.1 instead of Lemma C.7 of Jin et al. (2018). $\quad\square$

**Lemma F.3.** *(Bernstein concentration) For some given* $k, h, s, a \in [K] \times [H] \times \mathcal{S} \times \mathcal{A}$, *the following holds with probability at least* $1 - \delta$ *(with* $n = N_{k,h}(s,a)$),

$$
|\sum_{i=1}^n \alpha_n^i\left(r_{k_i,h} - R(s,a) + V_{h+1}^*(s_{k_i,h+1}) - P_{s,a}V_{h+1}^*\right)| \leq \sqrt{v_b(n,h,s,a)},
$$

*where*

$$
v_b(n,h,s,a) := \min\left\{c(\sqrt{\frac{H}{n+1}\cdot(\widehat{\mathbb{V}}_n\overline{V}_{h+1}(s,a) + H)\eta} + \frac{\sqrt{H^7SA\eta}\cdot\tau}{n+1}), 64\frac{H^3}{n+1}\right\}.
$$

*Proof.* Let $k_i$ denote the index of the episode when $(s,a)$ was visited for the $i_{\text{th}}$ time at step $h$. Set $x_i = \alpha_n^i(r_{k_i,h} - R_h(s,a))$, $y_i = \alpha_n^i(V(s_{k_{i+1}}) - P_{s_i,a_i}\cdot V)$ and consider filtration $\mathcal{F}_i$ as the $\sigma-$field generated by all random variables in the history set $\mathcal{H}_{k_i,h}$. We apply Azuma-Hoeffding (Lemma E.3) to calculate $|\sum_{i=1}^n x_i|$ and Azuma-Bernstein for $|\sum_{i=1}^n y_i|$. Consider with probability $1 - \delta$,

$$
\begin{aligned}
\left|\sum_{i=1}^n \alpha_n^i(V(s_{k_{i+1}}) - P_{s_i,a_i}\cdot V)\right| &\leq O(1)\cdot\left[\sum_{i=1}^n \sqrt{(\alpha_n^i)^2[\mathbb{V}_h V_{h+1}^*]_{s,a}\eta} + \frac{H^2\eta}{n}\right] \\
&\leq O(1)\cdot\left[\sum_{i=1}^n \sqrt{\frac{H}{n}[\mathbb{V}_h V_{h+1}^*]_{s,a}\eta} + \frac{H^2\eta}{n}\right].
\end{aligned}
$$

Using Lemma F.2 and a suitable union bound, we have with probability $1 - \delta$,

$$
\left| \sum_{i=1}^{n} \alpha_n^i (V(s_{k_{i+1}}) - P_{s_i,a_i} \cdot V) \right| \leq O(1) \cdot \left[ \sum_{i=1}^{n} \sqrt{\frac{H}{n+1} (\widehat{\mathbb{V}}_n \overline{V}_{h+1}(s,a) + H)\eta} + \frac{\sqrt{H^7 SA\eta} \cdot \chi}{n+1} \right]
$$
$$
\leq \sqrt{v_b(n,h,s,a)}.
$$

where the last term dominates the concentration of $\left| \sum_{i=1}^{n} x_i \right|$. $\qquad \square$

**Lemma F.4** (Based on Lemma C.1). *The samples from the posterior distributions and the mean of the posterior distributions as defined in Algorithm 3 satisfy the following properties: for any episode $k \in [K]$ and index $h \in [H]$,*

(a) *(Posterior distribution mean) For any given $s, a$, with probability at least $1 - \frac{2(k-1)\delta}{KH} - \frac{\delta}{KH}$,*

$$
\widehat{Q}_{k,h}(s,a) \geq Q_h^*(s,a) - \sqrt{v_b(n,h,s,a)}. \tag{48}
$$

(b) *(Posterior distribution sample) For any given $s, a$, with probability at least $p_1$ $(p_1 = \Phi(-1))$,*

$$
\tilde{Q}_{k,h}(s,a) \geq Q_h^*(s,a). \tag{49}
$$

(c) *(In Algorithm 2) With probability at least $1 - \frac{2k\delta}{KH}$,*

$$
\overline{V}_{k,h}(s_{k,h}) \geq V_h^*(s_{k,h}). \tag{50}
$$

*Proof.* The proof is identical to that of Lemma C.1 except that we use Lemma F.3 instead of Corollary D.2. $\qquad \square$

**Lemma F.5** (Based on Lemma C.6 of Jin et al. (2018)). *With probability at least $1 - 4\delta$, we have the following for all $k, h \in [K] \times [H]$,*

$$
\widehat{\mathbb{V}}_n \overline{V}_{k,h}(s_{k,h}, a_{k,h}) - \mathbb{V}_h V_{h+1}^{\pi_k}(s_{k,h}, a_{k,h}) \leq 2HP_{s_{k,h},a_{k,h}} \cdot (V_{h+1}^* - V_{h+1}^{\pi_k})
$$
$$
+ O\left( \frac{SA\sqrt{H^9 \chi^4}}{n} + \sqrt{\frac{H^7 SA \chi^2}{n}} \right),
$$

*where $n = N_{k,h}(s_{k,h}, a_{k,h})$ and $\chi = \log(JSAT/\delta)$.*

*Proof.* Consider,

$$
\widehat{\mathbb{V}}_n \overline{V}_{k,h}(s_{k,h}, a_{k,h}) - \mathbb{V}_h V_{h+1}^{\pi_k}(s_{k,h}, a_{k,h}) \leq \left| \widehat{\mathbb{V}}_n \overline{V}_{k,h}(s_{k,h}, a_{k,h}) - \mathbb{V}_h V_{h+1}^*(s_{k,h}, a_{k,h}) \right|
$$
$$
+ \left| \mathbb{V}_n V_{h+1}^*(s_{k,h}, a_{k,h}) - \mathbb{V}_h V_{h+1}^{\pi_k}(s_{k,h}, a_{k,h}) \right|,
$$

where for the first term we apply Lemma F.2 (which holds when Corollary F.1 and Lemma F.1 hold) and for the second term we have from the definition of variance,

$$
\left| \mathbb{V}_n V_{h+1}^*(s_{k,h}, a_{k,h}) - \mathbb{V}_h V_{h+1}^{\pi_k}(s_{k,h}, a_{k,h}) \right| \leq 2HP_{s_{k,h},a_{k,h}} \cdot (V_{h+1}^* - V_{h+1}^{\pi_k}).
$$

$\qquad \square$

