# OpenReview forum: "Q-learning with Posterior Sampling"
_ICLR.cc/2026/Conference — ICLR 2026 Poster_

### Official Review · Reviewer_7R7w · 2025-10-23

**Soundness:** 3
**Presentation:** 2
**Contribution:** 2
**Rating:** 2
**Confidence:** 3

**Summary:**

The paper proposes Posterior Sampling Q-Learning (PSQL), a tabular method that maintains Gaussian posteriors over Q-values and uses posterior samples for exploration; to obtain a $\tilde{O}(H^{2}SAT^{1/2})$ regret bound, the analyzed variant makes the bootstrapped target optimistic by taking the maximum over multiple posterior draws for the next-state value. While the vanilla single-sample version exhibits strong empirical performance, the optimistic variant required for the theory is not competitive, lagging behind RLSVI and offering no gains over Staged-RandQL on the authors’ own benchmarks.

**Strengths:**

- Provides a near-optimal $\tilde{O}(H^{2}SAT^{1/2})$ regret analysis for a posterior-sampling variant of tabular Q-learning.
- The vanilla practical instantiation consistently outperforms classical baselines such as UCB-QL and Staged-RandQL on the reported tasks, showing promise for posterior-sampling driven exploration in tabular settings.

**Weaknesses:**

- The empirically evaluated “vanilla” algorithm differs from the theoretically analyzed optimistic variant; the latter is not competitive with RLSVI or Staged-RandQL on the reported benchmarks, leaving the practical relevance of the theory unclear. This mismatch makes it difficult to understand the paper’s positioning: the theoretically grounded algorithm underperforms in practice and does not improve existing regret bounds, yet the practical algorithm—which shows promise—is not benchmarked against other practical exploration baselines on more demanding tasks.
- Experimental coverage is very limited (two toy tabular environments) and omits stronger practical baselines (e.g., recent posterior-sampling or optimistic methods); if the practical implementation departs from the analyzed algorithm, broader comparisons to other exploration strategies become essential, but they are absent here.
- Presentation can be clarified: the introduction currently reads as an unstructured list of related work rather than positioning the contribution, and some statements about model-based methods could be refined.
- The “first Bayesian posterior” claim is confusing: several prior works already use posterior-sampling within Q-learning, so the novelty of this phrasing is unclear and should be clarified.

**Questions:**

- Given that the analyzed optimistic variant performs worse than RLSVI and Staged-RandQL, how do the authors reconcile the theoretical guarantees with practical usefulness? Can they provide intuition or evidence that the analyzed algorithm offers benefits beyond these baselines?
- If the practical PSQL* deviates from the analyzed variant, do the authors plan to benchmark it against more practical exploration algorithms such as [1] on more challenging domains (e.g., Atari, Maze2D) to establish empirical competitiveness?
- In lines 118–119 the paper says model-based methods directly model the rewards and transitions “instead of” the implied value function or policies; could the authors clarify this phrasing? In practice, model-based RL often uses learned models to improve the value function and/or policy.
- What exactly are the authors claiming as “first” in the first contribution point, “provided by the Bayesian posterior”? Could they clarify how their notion of “Bayesian posterior” differs from earlier posterior-based Q-learning efforts (e.g., [1]) and what concrete novelty they intend to highlight?

[1] Ishfaq et al., Provable and Practical: Efficient Exploration in Reinforcement Learning via Langevin Monte Carlo, ICLR 2024.

---

> ### Author Response · Authors · 2025-11-17
> **Thank you for your careful review**
>
> We thank the reviewer for the helpful comments and for carefully reading our paper. ***Please also look at the public comment and our response at the top, where we elaborate on multiple points about novelty in our work and relationship to LSVI-like algorithms***
>
> ## **Scope: model-free Q-learning vs. model-based methods.**
>
>  The main theme of the paper is to study theoretical guarantees for model-free algorithms for TD-learning algorithms, specifically Q-learning–type methods that do not explicitly estimate transition kernels or store all past data in a structured replay buffer. We are specifically referring to Q-learning as discussed in [2,3]. In contrast, algorithms such as RLSVI and LSVI (e.g., the variant in [1] ) are effectively model-based: they maintain and fit on a large dataset of past transitions, which is known (e.g., Tiapkin et al.) to yield stronger convergence/sample-complexity guarantees. While model-based methods are more sample-efficient, model-free algorithms like Q-learning (TD-learning-based algorithm) remain attractive due to their simplicity and scalability. Consistent with this picture, PSQL is somewhat inferior to RLSVI but comparable to Staged-RandQL in our experiments. Further PSQL is much simpler to describe and implement as compared to Staged-RandQL [4]
>
>
> ## **Theoretical focus and experimental scope.**
>  The main contribution of the paper is theoretical and methodological: we introduce a posterior-sampling variant of Q-learning (Temporal Difference-learning based algorithm) and derive regret guarantees for it. A comprehensive large-scale empirical study of PSQL* is beyond the scope of the current work and is a natural direction for future research.
>
>
> ## **Clarification of model-free vs. model-based phrasing (Lines 118–119).**
>  We will clarify Lines 118–119. The intended distinction is that model-free algorithms like Q-learning update Q-values via incremental TD/Bellman updates using the current transition sample and do not maintain an explicit transition model or a global dataset used for solving a regression or planning problem at each step. In contrast, model-based algorithms such as UCBVI and LSVI/RLSVI either (i) maintain empirical transition models or (ii) store past data and repeatedly solve a regression/planning subproblem from this dataset, i.e., there is ***no*** Temporal Difference-based update. We will make this nuanced distinction precise in the revised version.
>
>
> ## **Novelty of posterior sampling with Q-learning.**
>
>  Our algorithm is, to the best of our knowledge, the first to motivate posterior sampling directly within a Q-learning–style (TD-learning based) update by maintaining an explicit posterior distribution over the Q-function itself. In our setting, the recursive Bellman update and its uncertainty drive the posterior evolution of Q function. Infact, our approach is the first posterior sampling approach for any TD-learning based algorithm with theoretical guarantees. This is conceptually different from RLSVI/LSVI-type methods, which fit a Q-function (or its parameters) to a dataset of past transitions via regression and then act greedily with respect to that fitted function. We will clarify this distinction in the manuscript.
>
> >[1] Ishfaq et al., Provable and Practical: Efficient Exploration in Reinforcement Learning via Langevin Monte Carlo, ICLR 2024.
>
> >[2] Christopher JCH Watkins and Peter Dayan. Q-learning. Machine learning, 8:279–292, 1992.
>
> >[3] Chi Jin, Zeyuan Allen-Zhu, Sebastien Bubeck, and Michael I Jordan. Is Q-learning provably efficient?Advances in neural information processing systems, 31, 2018.
>
> >[4] Daniil Tiapkin, Denis Belomestny, Daniele Calandriello, Eric Moulines, Remi Munos, Alexey
> Naumov, Pierre Perrault, Michal Valko, and Pierre Ménard. Model-free posterior sampling via
> learning rate randomization. Advances in Neural Information Processing Systems, 36:73719–73774, 2023.

---

> ### Author Response · Authors · 2025-11-27
>
> Dear Reviewer 7R7w,
>
> Thank you again for taking the time to read our paper. We hope our rebuttal helps clarify your main points. We try to highlight our main novelty and the comparison to LSVI/RLSVI-type methods. If you have the opportunity to revisit the paper and our additional responses (including the public comment at the top), we would appreciate any updated feedback.

---

### Official Review · Reviewer_wHwf · 2025-10-25

**Soundness:** 3
**Presentation:** 2
**Contribution:** 2
**Rating:** 6
**Confidence:** 3

**Summary:**

The authors proposed a model-free exploration method based on introducing the idea of Randomized Least-Squares Value Iteration (RLSVI) into the Q-learning framework. This algorithm, called Posterior Sampling Q-learning (PSQL), uses a Gaussian posterior estimate of Q-values and achieves a regret guarantee of $\tilde{O}(H^2 \sqrt{SAT})$, which is the same as Q-learning with UCB bonuses or Staged Randomized Q-learning. Additionally, the algorithm shows strong empirical performance on standard low-dimensional benchmarks.

**Strengths:**

- Interesting alternative explanation of the UCB-Q-learning learning rate, that appears from the additional entropy regularization in the variational approximation, with a clear intuition of "collapse avoidance" with entropy due to bias in the estimate;
- Strong empirical performance as well as theoretical regret guarantees;

**Weaknesses:**

- Lack of empirical comparison with a usual RandQL. Although this method does not offer the same rigorous guarantees as its staged version, it would be interesting to compare PSQL* and a usual RandQL without stages.
- The regret bound does not match the regret bound of a variance-reduced version of Q-learning (Li et al. 2021);

**Questions:**

- It would be beneficial to discuss an RLSVI-style model-based algorithm that achieves the minimax optimal regret guarantee (Xiong et al. 2022).
- What prevents you from using a standard RSLVI analysis with a single sample there?
- What prevents you from extending the sketch of the proof in Appendix F to a complete proof for a variance-shaped noise? What is a main challenge there?
- Is it possible to provide a deep-learning version of your method?

### References

Xiong, Z., Shen, R., Cui, Q., Fazel, M., & Du, S. S. (2022). Near-optimal randomized exploration for tabular Markov decision processes. Advances in neural information processing systems, 35, 6358-6371.

---

> ### Author Response · Authors · 2025-11-17
> **Thank you for your constructive review**
>
> We appreciate your time and constructive review.
>
> ## **Comparison with RandQL.**
>  In our toy experiments, PSQL\* outperforms RLSVI, while RandQL is inferior to RLSVI [1]. Thus, PSQL\* is indirectly stronger than RandQL in this setup. We will clarify this comparison more explicitly in the revised version.
>
>
> ## **Model-free Q-learning vs. RLSVI/LSVI.**
>  The main theme of the paper is to study theoretical guarantees for model-free TD-learning algorithms, specifically Q-learning–type methods that do not explicitly estimate transition kernels or store all past data in a structured replay buffer (in the sense of Q-learning analyzed in [2,3]). In contrast, RLSVI/LSVI-type methods [e.g., 4] are effectively model-based: they estimate transition dynamics or store past data and solve regression/planning problems, rather than relying purely on TD updates. In this sense, Q-learning and RLSVI are fundamentally different algorithms. We will add this clarification to the paper.
>
>
> ## **Appendix F and posterior sampling.**
>  One of our motivations in Section 3.1 is to provide an explicit posterior distribution over the Q-function, unlike RLSVI-type methods. In Appendix F, although we obtain a tighter regret guarantee within that framework, the resulting algorithm is not a genuine posterior sampling method: the variance updates are not derived from Bayes’ rule or an explicit ELBO objective. The regret bound is complete for a randomized algorithm, but the procedure is not a clean posterior sampling algorithm. We will make this distinction clearer.
>
>
> ## **Deep RL / function approximation.**
>  It may be possible to design heuristic extensions of PSQL that work with neural networks, but we do not yet see an obvious formulation that supports rigorous regret guarantees in the deep RL setting. We view this as important future work, which falls beyond the scope of the current paper.
>
>
>
>
> >[1] Daniil Tiapkin, Denis Belomestny, Daniele Calandriello, Eric Moulines, Remi Munos, Alexey
> Naumov, Pierre Perrault, Michal Valko, and Pierre Ménard. Model-free posterior sampling via
> learning rate randomization. Advances in Neural Information Processing Systems, 36:73719–73774, 2023.
>
> >[2] Christopher JCH Watkins and Peter Dayan. Q-learning. Machine learning, 8:279–292, 1992.
>
> >[3] Chi Jin, Zeyuan Allen-Zhu, Sebastien Bubeck, and Michael I Jordan. Is Q-learning provably efficient?Advances in neural information processing systems, 31, 2018.
>
> >[4] Xiong, Z., Shen, R., Cui, Q., Fazel, M., & Du, S. S. (2022). Near-optimal randomized exploration for tabular Markov decision processes. Advances in neural information processing systems, 35, 6358-6371.

---

> ### Author Response · Authors · 2025-11-27
>
> Dear Reviewer wHwf,
>
> Thank you again for taking the time to read our paper. We hope our rebuttal helps clarify your main concerns. If you have the opportunity to revisit the paper and our additional responses (including those to other reviewers), we would be very grateful for any updated feedback.

---

> > ### Comment · Reviewer_wHwf · 2025-11-27
> >
> > Thank you very much for your answer, and thank you very much for the clarification regarding the algorithm presented in Appendix F. I would like to maintain my score.

---

### Official Review · Reviewer_UBp9 · 2025-10-27

**Soundness:** 3
**Presentation:** 3
**Contribution:** 3
**Rating:** 8
**Confidence:** 4

**Summary:**

This paper presents Q-Learning with Posterior Sampling (PSQL), a model-free reinforcement learning algorithm that employs Gaussian posteriors on Q-values for exploration. By interpreting Q-learning as a Bayesian inference problem with a regularized ELBO objective, the authors design a conceptually simple algorithm that combines single-sample posterior sampling for action selection with multiple-sample optimism for target computation. This design allows them to prove a near-optimal regret bound of O(H^2 \sqrt(SAT)), while preliminary experiments show competitive or superior empirical performance compared to UCBQL, RLSVI, and Staged-RandQL.

**Strengths:**

The work is theoretically grounded, algorithmically simple, and provides new insights into the Bayesian interpretation of Q-learning. The regret guarantee is strong, and the analysis tackles key challenges in combining posterior sampling with TD learning.

**Weaknesses:**

Using Gaussian posteriors on Q-values may destroy important structural properties of Q-functions (e.g., boundedness or Bellman consistency), since Gaussian distributions are unbounded. The choice of posterior variance is subtle and strongly affects performance, requiring careful tuning. Moreover, the use of multiple posterior samples for target computation increases the algorithm’s computational complexity, and the theoretically unanalyzed single-sample variant (PSQL*) outperforms the analyzed one in practice, indicating a gap between theory and implementation.

**Questions:**

1. Can alternative posterior distributions preserve Q-value structure more faithfully than Gaussians?

2. Is there a principled or adaptive way to select the posterior variance to avoid heuristic tuning?

3. Can the multiple-sampling step for optimism be replaced with a cheaper or more elegant alternative?

4. How does the approach scale to function approximation or deep RL settings?

5. How does posterior optimism interact with TD bootstrapping bias in long horizons?

---

> ### Author Response · Authors · 2025-11-17
> **Thank you for your constructive review**
>
> We thank the reviewer for the thoughtful and constructive comments.
>
> ## **On Gaussian priors/likelihood and the Bellman equation.**
>
>  Respectfully, we do not agree that the choice of a Gaussian likelihood and prior is overly restrictive or interferes with the Bellman equation. Our goal is to give a Bayesian interpretation and methodology for a traditionally frequentist Q-learning algorithm [2,3]. As is standard in Bayesian statistics, we maintain a distribution over the quantities we wish to estimate. As learning progresses, the posterior concentrates (its variance decreases) around its mean, which approaches the optimal Q-function that is bounded and satisfies the optimal Bellman equation. Importantly, the Bellman equation is a property of the optimal Q-function only; no sub-optimal estimate can satisfy it exactly.
>
>
> ## **On entropy-based regularization and the posterior variance.**
>  We view the entropy-based regularization in Eq. (8) (and its solution in Lemma B.1) as a principled way to select the posterior variance. This is, in fact, one of the contributions of our work: it provides an alternative, principled method for choosing or estimating the variance compared to earlier Q-learning algorithms with UCB bonuses [2].
>
>
> ## **On PSQL\* as an alternative approach.**
>  The alternative approach is PSQL* which we also implement in our experiments. A theoretical analysis of PSQL* remains an open problem, and we explicitly consider it a central direction for future works.
>
>
> ## **On function approximation and deep RL.**
>
>  Extending PSQL to function approximation or deep RL settings is a promising and natural direction; however, it introduces additional challenges (representation, stability, and approximation error) that are orthogonal to our current focus. We therefore regard such extensions as fruitful areas for future work, beyond the scope of this paper.
>
>
> ## **On TD bootstrapping bias.**
>  TD bootstrapping bias is an important design consideration for our algorithm. As discussed in Section 4.1, the multiple-sampling variant of PSQL is specifically introduced to address this issue: by making the TD target estimate optimistic, it prevents the bias from compounding multiplicatively over time.
>
>
> >[1] Christopher JCH Watkins and Peter Dayan. Q-learning. Machine learning, 8:279–292, 1992.
>
> >[2] Chi Jin, Zeyuan Allen-Zhu, Sebastien Bubeck, and Michael I Jordan. Is Q-learning provably efficient?Advances in neural information processing systems, 31, 2018.

---

> > ### Comment · Reviewer_UBp9 · 2025-11-28
> >
> > Thank you  for your answers and clarifications. I still believe that prior Gaussian distribution on Q values is counterintuitive  as it, for example, can destroy their positivity.  Still I maintain my high score.

---

> ### Author Response · Authors · 2025-11-27
>
> Dear Reviewer UBp9,
>
> Thank you again for taking the time to read our paper. We hope our rebuttal helps clarify your main concerns. If you have the opportunity to revisit the paper and our additional responses (including those to other reviewers), we would be very grateful for any updated feedback.

---

### Official Review · Reviewer_ZF74 · 2025-10-31

**Soundness:** 2
**Presentation:** 3
**Contribution:** 2
**Rating:** 2
**Confidence:** 3

**Summary:**

The authors introduce PSQL, a novel model-free method that utilizes Gaussian Bayesian inference on Q-values. This approach incorporates a different target value based on the optimism principle, though this optimistic target value does not influence the actual decision-making process. The paper provides regret bounds that are nearly optimal and comparable to those achieved by other posterior sampling methods. Additionally, the authors establish a modified version, PSQL, which uses the target value chosen as in standard Q-learning. The results demonstrate that PSQL outperforms several baseline methods in tabular environments.

**Strengths:**

Authors discuss the limitations of the analysis of the vanilla PSQL algorithm

**Weaknesses:**

- In my opinion, the empirical results are not sufficiently extensive. It would be interesting, for example, to consider a comparison with the PSRL algorithm, which was shown to outperform Staged-RandQL in a recent study (Tiapkin et al., 2023). Furthermore, the paper lacks a comparison in more complex environments, specifically those with a continuous state space;
- Another interesting direction would be to extend this algorithm to more practical scenarios with a general state space. If this is possible, what quantity should be chosen for the variance in that setting?
- What regret bound can be achieved in the "vanilla version" of the PSQL algorithm? Could a non-trivial polynomial bound be established
- It appears there is a potential issue with the inequality  $\mathbb{E}[\tilde{X}^{\text{alt}}| \bar{\mathcal{O}}^{\text{alt}}] \leq \underline{X}$ in Lemma E.1. This is because the statement assumes $\mathbb{E}[\tilde{X}]\geq \underline{X}$, but the inequality seems to require the opposite for the proof to hold.

**Questions:**

See weaknesses.

**Details Of Ethics Concerns:**

-

---

> ### Author Response · Authors · 2025-11-17
> **Thank you for your review**
>
> We thank the reviewer for the detailed and constructive feedback.
>
> ## **Scope and main goal.**
>
> Our main goal is to propose a new posterior sampling algorithm that is compatible with TD-learning methods such as Q-learning and to rigorously analyze its theoretical performance. The focus of the paper is on establishing a regret guarantee for this class of model-free posterior sampling algorithms.
>
>
> ## **Role of the experiments.**
> The experiments are intentionally conducted in tabular / toy settings to motivate and isolate the core question of posterior sampling vs. UCB in model-free RL. We agree that experiments in continuous action spaces and more complex environments would further promote large-scale adoption of PSQL, but we view these as beyond the scope of the current, theory-focused paper.
>
>
> ## **On PSRL and model-based methods.**
> By design, PSQL is a model-free algorithm: it does not maintain estimates of transition dynamics or posteriors over them. In contrast, PSRL is a model-based method. As shown in Tiapkin et al., 2023 and confirmed in our own experiments, model-based methods can be more sample efficient but are generally harder to scale. Since this work is focused on model-free algorithms, we did not include PSRL as a baseline.
>
>
> ## **Extensions beyond the tabular setting.**
>  It is currently an open question how to compute principled UCB bonuses or posterior variances for general (non-tabular) model-free Q-learning algorithms. Some existing approaches obtain such quantities either via experience replay or by adopting a model-based perspective, but a fully general, model-free treatment remains unresolved. We will clarify this in the discussion.
>
>
> ## **Regret of vanilla PSQL (PSQL\*).**
>  The theoretical analysis in the paper applies to the optimistic variant of PSQL. Deriving a regret bound for vanilla PSQL* remains an open problem. Given its strong empirical performance, we believe that new analytical tools may allow one to establish non-trivial guarantees for PSQL*, and we explicitly state this as an interesting direction for future work in our conclusion section.
>
>
> ## **Typo in Lemma E.1.**
>  We thank the reviewer for catching the typo in line 1328 (Lemma E.1 proof and Eq. (34)). We have corrected it in the revised version; the logic of the proof remains unchanged.

---

> ### Author Response · Authors · 2025-11-27
>
> Dear Reviewer ZF74,
>
> Thank you again for taking the time to read our paper. We hope our rebuttal helps clarify your main concerns, especially regarding the goal and scope of our work and the role of the experiments in relation to prior literature. If you have the opportunity to revisit the paper and our additional responses (including those to other reviewers), we would be very grateful for any updated feedback.

---

### Public Comment · ~Haque_Ishfaq1 · 2025-11-16
**Missing related work and comparison against existing posterior-sampling based Q-learning algorithms**

Dear Authors,

We would like to point out some highly related previous works that the authors failed to mention and discuss which can be seen as efforts to design posterior-sampling based/ Bayesian Q-learning algorithms.

In Line 224, the authors claim that "We first present a Bayesian posterior-based derivation of the Q-learning update rule". However, [1], [2], [3] -- all of them can be seen as posterior based Q-learning algorithm. We thank the `Reviewer 7R7w` for pointing out [2] as an earlier effort in this direction. We would appreciate if the authors could acknowledge and compare against these earlier efforts in their submission.

In Line 201-203, the authors say that:

> There have been relatively limited studies on model-free, sample-efficient and computationally efficient Bayesian algorithms. Dann et al. (2021) proposed one such framework but is computationally intractable. Our work aims to fill this gap.

However, in [3], we provided a tractable and provably efficient version of Feel-Good Thompson Sampling algorithm along with thorough experiments in Atari and `N-Chain` environments.

In Line 373, the authors discuss mutliple sampling techniques. Multiple sampling for gaining optimism is also used in LSVI-PHE in [1]. Could you please discuss how the approach presented here differs from that of LSVI-PHE?


[1] Ishfaq et al., Randomized Exploration for Reinforcement Learning with General Value Function Approximation, ICML 2021

[2] Ishfaq et al., Provable and Practical: Efficient Exploration in Reinforcement Learning via Langevin Monte Carlo, ICLR 2024

[3] Ishfaq et al., More Efficient Randomized Exploration for Reinforcement Learning via Approximate Sampling, RLC 2024

---

> ### Author Response · Authors · 2025-11-17
> **Thanks for raising an interesting point**
>
> Hi Haque,
>
> Thank you very much for your thoughtful comment and for highlighting these related works. We also refer readers to our rebuttal to *Reviewer 7R7w*, where we elaborate on some of these points.
>
> 1. We would like to clarify the distinction we are drawing between LSVI-based algorithms (such as those in [1, 2, 3], as well as RLSVI/RLSVI-like methods) and the classical Q-learning–type algorithms that are the focus of our paper. In our work, we use “Q-learning” to refer specifically to TD-learning–based methods where new estimates are obtained by bootstrapping from previous Q-estimates. As we discuss in Section 4.1 of our submission, this bootstrapping nature makes posterior sampling challenging, because the targets themselves depend on the current, uncertain Q-function. In contrast, LSVI-based algorithms do not suffer from this particular TD bootstrapping issue: they rely on solving regression or planning problems over stored data, rather than updating purely via TD-style bootstraps. A related nuance is that our focus is on model-free algorithms, whereas LSVI-type methods (including [1, 2, 3]) are effectively model-based or data-reuse–based in the sense that they store and repeatedly process a growing dataset of transitions. We have cited your papers in the relevant sections in the updated submission.
>
> 2. Within this Q-learning / TD-learning perspective, our contribution is, to the best of our knowledge, the first to carry out an explicit (approximate) posterior-sampling construction for a TD-based Q-learning update, with an explicit posterior distribution over the Q-function and an associated regret analysis. We fully acknowledge that your work [3] provides a very strong and practically important baseline; however, our understanding is that the main algorithm in [3] is not model-free in the sense we adopt here. For example, in the tabular setting the storage requirement scales on the order of $\Omega(S^2A)$, whereas our focus is on algorithms whose memory scales more like classical tabular Q-learning. We try to emphasize these themes, model-free, TD-based updates, and tabular Q-learning in several places in the paper.
> For this reason, we respectfully disagree with the statement that ***“[1], [2], [3] – all of them can be seen as posterior-based Q-learning algorithms”*** when Q-learning is interpreted as a TD-learning algorithm with bootstrapping. Under that broader usage, many value-based methods such as RLSVI or tabular LSVI (e.g., UCB-LSVI) could also be labeled “Q-learning-like,” which blurs the specific TD-learning focus that our work is trying to address.
>
> 3. Finally, regarding multiple sampling: in our algorithm (Section 4.1), multiple sampling is used to make the TD targets optimistic, rather than to make the Q-function optimistic itself. This is the primary conceptual difference we observe in relation to *LSVI–PHE*-style multiple sampling. That said, we fully agree that your work provides another important example (among others we cite) of using multiple sampling to induce optimism in reinforcement learning.
>
> We again thank you for the detailed feedback and for pointing us to these closely related contributions.

---

> > ### Public Comment · ~Haque_Ishfaq1 · 2025-11-20
> > **Thanks for clarification**
> >
> > Dear Authors,
> >
> > Thanks for your response and clarifying the nuanced differences.  It would be great if you could incorporate some of these points and nuanced differences in the paper. Especially how the submission differs from the LSVI based methods due to the focus on TD learning based methods.

---

> > > ### Author Response · Authors · 2025-11-20
> > >
> > > Thanks for your comments
> > > We updated the references already. We will add a brief discussion in the related works section after the rebuttal period.

---

### Meta-Review · Area_Chair_ULxU · 2026-01-07

**Summary:**

The paper introduces Q-Learning with Posterior Sampling (PSQL), a model-free reinforcement learning algorithm that leverages Gaussian posteriors for exploration. The authors establish a near-optimal regret bound of \tilde{O}(H^2\sqrt{SAT}) in the tabular episodic setting, matching known lower bounds. A key theoretical contribution is the first regret analysis of a TD-learning–based posterior sampling algorithm, distinguishing this work from prior analyses that focus on least-squares value iteration–based methods. After clarification during the rebuttal, reviewers largely viewed the work as theoretically solid and well-founded. Remaining concerns are relatively minor on the practical relevance of the specific algorithmic variant analyzed and the limited scope of the empirical evaluation.

**Reviewer Concerns:**

Reviewers were initially confused about the novelty of the algorithmic contribution of this paper compared with RLSVI-type work (e.g., RLSVI, LSVI-PHE, LMC-LSVI), which also introduces posterior sampling into reinforcement learning. The authors provided a more nuanced distinction between TD-based Q-learning and LSVI-type methods, emphasizing that the current paper focuses on TD learning with bootstrapping, whereas prior work is essentially model-based in the sense that it stores and regresses over past experience during learning. It is worth noting, however, that this “model-based” characterization primarily applies to the linear MDP versions of these algorithms. Several of the cited works also include simple and practical implementations that rely only on SGD/SGLD-style updates to learn value functions while injecting randomized exploration; see, for example, Provable and Practical: Efficient Exploration in Reinforcement Learning via Langevin Monte Carlo (ICLR 2024).

Several minor concerns remain regarding the discrepancy between the algorithms analyzed in the theoretical section and those used in the empirical evaluation. Reviewers pointed out that the theoretically analyzed “optimistic” variant required to obtain the regret bound is not competitive with existing methods such as RLSVI on the authors’ own benchmarks. In addition, the experimental scope is limited to two toy tabular environments, with no evaluation on more complex or continuous state-space benchmarks. There are also outstanding concerns regarding the use of Gaussian distributions for bounded Q-values and the heuristic nature of posterior variance tuning.

Overall, from the area chair's understanding, the theoretical contribution of this paper is novel and clearly differentiated from existing literature. Given that **the paper is primarily theoretical in nature and that empirical evaluation is not its main focus, it is recommended for acceptance**. Nevertheless, the authors are strongly encouraged to improve the presentation and clarity of their theoretical contributions relative to existing work. For the empirical validation, the authors should also **tone down claims regarding connections to successful modern deep RL algorithms**. In particular, LMC-LSVI–type approaches are more closely aligned with empirical DQN implementations, which use (nonlinear) regression to approximate Q-values and have demonstrated the empirical success of posterior sampling in DQN-style methods.

**Reviewer Scores:**

Reviewer UBp9: Score: 8. They remained supportive due to the theoretical grounding but still noted the gap between theory and implementation.

Reviewer wHwf: Score: 6. They maintained this score but expressed that they would not mind if the paper was rejected.

Reviewer ZF74: Score: 2. They felt the empirical results were insufficient and lacked practical scalability demonstrations. Based on the detailed rebuttal, their score might increase to 4.

Reviewer 7R7w: Score: 2. Their primary concern remained the mismatch between the practical PSQL* and the analyzed variant, which underperforms previous algorithms. Their score might increase due to the focus on the novel theoretical contribution of this paper.

---

### Decision · Program_Chairs · 2026-01-26

Accept (Poster)